# CAN EXPLORATION SAVE US FROM ADVERSARIAL ATTACKS? A REINFORCEMENT LEARNING APPROACH TO ADVERSARIAL ROBUSTNESS

## ABSTRACT

Although considerable progress has been made toward enhancing the robustness of deep neural networks (DNNs), they continue to exhibit significant vulnerability to gradient-based adversarial attacks in supervised learning (SL) settings, which necessitates the need for alternative theoretical foundations for robustness. To address this gap, we investigate adversarial robustness under reinforcement learning (RL), training image classifiers with policy-gradient objectives and $\epsilon$-greedy exploration. When training models with several architectures on CIFAR-10, CIFAR-100, and ImageNet-100 datasets, RL consistently improves adversarial accuracy under white-box gradient-based attacks. Our results show that on a representative 6-layer CNN, adversarial accuracy increases from approximately 5% to 55% on CIFAR-10, 2% to 25% on CIFAR-100, and 5% to 18% on ImageNet-100, while clean accuracy decreases only 3–5% relative to SL. However, transfer analysis reveals that adversarial examples crafted on RL models transfer poorly: both SL and RL retain approximately 43% accuracy against these attacks. In contrast, adversarial examples crafted on SL models transfer effectively, reducing both SL and plain RL to around 8% accuracy. This indicates that while plain RL can prevent the generation of strong adversarial examples, it remains vulnerable to transferred attacks from other models, thus requiring adversarial training (RL-adv, $\sim$30% adversarial accuracy) for comprehensive defense against cross-model attacks. Analysis of loss geometry and gradient dynamics shows that RL induces smaller gradient norms and rapidly changing input-gradient directions, reducing exploitable information for gradient-based attackers. Despite higher computational overhead, these findings suggest RL-based training can complement existing defenses by naturally smoothing loss landscapes. Overall, this work proposes a hypothesis, backed by empirical evidence, describing a mechanism for improving robustness, motivating future research on hybrid methods that combine SL efficiency with RL-driven gradient regularization.

## 1 INTRODUCTION

As artificial intelligence (AI) is increasing in power, a growing number of users actively or passively use AI technologies daily. [1] Consequently, ensuring the security of AI systems has therefore become critical, as numerous studies have revealed notable security vulnerabilities, for example (Szegedy et al., 2014; Goodfellow et al., 2014; Eykholt et al., 2018; Biggio & Roli, 2018). A neural network is a fundamental component of AI, and machine learning (ML) algorithms provide the methodology to optimize its performance. One of the most important vulnerabilities in ML arises from adversarial attacks, which exploit the gradient-based optimization at the core of supervised learning (SL). This attack can subtly manipulate a model's decisions in ways that are imperceptible to humans (Goodfellow et al., 2014; Eykholt et al., 2018). To defend against adversarial attacks, various strategies have been proposed to enhance the robustness of neural networks, including noise injection during training, data augmentation, and adversarial training (Bishop, 1995; Cohen et al., 2019; Hendrycks

---

[1]*Artifacts:* anonymous code and configuration files are available at https://github.com/iclr2026aerl/ICLR2026-AERL. *LLM Usage:* we disclose our use of large language models in Appendix A.10.

et al., 2020; Zhang et al., 2018; Madry et al., 2018; Kurakin et al., 2017). However, several studies have shown that even these robust models can be compromised if an adaptive attacker has access to robust models (Aghabagherloo et al., 2023; He et al., 2017; Aghabagherloo et al., 2025b).

Reinforcement learning (RL), another core category in ML, is widely used in control systems, robotics, and other sequential decision-making tasks to improve performance and robustness (Mnih et al., 2015; Levine et al., 2016; Pinto et al., 2017; Akhtar & Mian, 2018; Biggio & Roli, 2018). Nevertheless, RL-based approaches to improve robustness for classification tasks remain comparatively underexplored. **Our hypothesis** is that RL can enhance model robustness compared to SL under gradient-based adversarial attacks (e.g., projected gradient descent (PGD)). RL's property of exploration and policy optimization does not rely on explicit end-to-end input gradients (e.g., via black-box policy search). Intuitively, for both standard and "robustified" models trained by gradient-based optimization algorithms (SL), attackers who recover reliable gradients can perform highly effective attacks; however, RL-style optimizations may induce flatter gradients that are harder to exploit with gradient-based attacks.

Most adversarial robustness work for image classification builds on SL objectives such as empirical risk minimization or min–max adversarial training (Madry et al., 2018). While effective, these pipelines can still lead models to rely on non-robust yet predictive cues that are easy to exploit (Ilyas et al., 2019). We therefore investigate whether treating classification as decision-making over a short sequence can help. Concretely, we study an RL-based classifier: an agent that processes an input, receives a task reward for correct decisions, and is trained to keep its decisions stable when the input is slightly perturbed. In this view, adversarial perturbations play the role of external disturbances. Training with simple stability penalties and worst-case exposure can then encourage more reliable decisions. In this paper, we instantiate such a classifier and compare it with strong supervised baselines under the same attack budgets and training time.

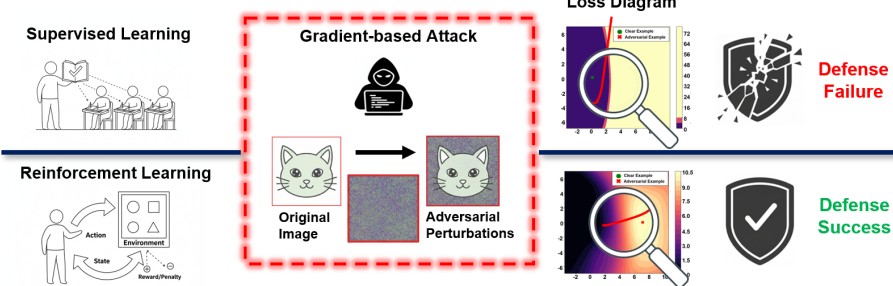

Figure 1: We compare **SL** and **RL** training for image classification under gradient-based adversarial attacks. **Top (SL):** SL-trained models retain sharper, more informative gradients, which adversaries can readily exploit. **Bottom (RL):** RL-trained models exhibit flatter, less informative gradients, offering no clear gradient direction for attack. This mechanism matches our empirical results on CIFAR-10/100 and ImageNet-100.

Our goal is to employ RL in a classification task to make the model robust against adversarial examples (AEs). Our primary results indicate that RL-trained models can withstand adversarial attacks more effectively than SL-trained models. Furthermore, our empirical results lead to a fundamental hypothesis explaining why an RL-based classifier can be more robust than an SL-based classifier. This hypothesis, supported by empirical evidence, describes a potential mechanism for improving robustness and motivates future research on hybrid methods that combine the efficiency of SL with RL-driven gradient regularization. **Our main contributions**: (i) An exploratory analysis of the effect of employing RL in a classification task to make the model robust against AEs; (ii) Experimental results demonstrating our claimed robustness on CIFAR-10, CIFAR-100, and ImageNet-100; (iii) Theoretical explanation of why RL-based models are more robust.

## 2 RELATED WORK

Machine learning is widely used in AI tasks, including SL, unsupervised learning, and reinforcement learning. In classification tasks, deep neural network (DNN) image classifiers under SL effectively

learn discriminative patterns, and surpass human performance on several vision tasks (He et al., 2015). (Additional SL/RL training details are provided in Appendix A.5 and Appendix A.6)

Despite the mentioned capability of DNN-based image classifiers, they show susceptibility to a wide range of attacks (Ozdag, 2018). DNNs are susceptible to privacy and security threats such as (i) data poisoning (Zhao et al., 2025), where adversaries inject poisoned data into the training set to corrupt the model, (ii) evasion (also known as adversarial attacks), where input samples are intentionally perturbed in a way that causes the model to misclassify them during the testing phase, (iii) model inversion, where the goal is feature reconstruction of samples from the training set using the model's outputs (Fredrikson et al., 2015), (iv) membership inference attacks, which attempt to determine whether a specific data sample was part of the model's training dataset or not, etc (Shokri et al., 2017). From all these attacks, evasion attacks are the most well-known ones against DNNs (Ilyas et al., 2019). These types of attacks can be considered especially practical because they exploit non-robust features of the data, meaning that even naturally occurring perturbations in input can sometimes lead to similar misclassification behavior (Ilyas et al., 2019).

## 2.1 Adversarial Attacks on Classification Task

The perturbations to generate adversarial attacks can be perceptible (Schneider & Apruzzese, 2023), where the attacker aims to deceive both humans and DNNs, or imperceptible to the human eyes (Aghabagherloo et al., 2025a; Ilyas et al., 2019). In most adversarial scenarios, the perturbations are intentionally imperceptible, as the primary objective is mainly to mislead the DNN without introducing noticeable changes to human observers.

Adversarial attacks on DNNs are classified as (i) white-box attacks, where adversaries have complete knowledge of the trained model, and (ii) black-box attacks, where the attacker lacks complete knowledge of the learned model's parameters. In Zeroth Order Optimization (ZOO) (Chen et al., 2017), a widely recognized black-box attack, the attacker has only access to the input data and the output of the model. Among white-box attacks, the Carlini & Wagner (C&W) attack, Projected Gradient Descent (PGD), and Fast Gradient Sign Method (FGSM) are the most widely studied methods. To defend against attacks, several studies (Wu et al., 2023) tried to robustify the DNNs, while others have demonstrated the weaknesses of current defenses. This has become a cycle of articles demonstrating DNNs' vulnerabilities, proposing robustification methods, and bypassing those robustifications. This is especially evident when the attacker has access to the robustification method (Aghabagherloo et al., 2023; Athalye et al., 2018). These works showed that even when a model is robust, an adversary aware of the robustification approach can still successfully generate attacks.

## 2.2 Robustness and Reinforcement Learning

Although RL's contribution to robustness has been rarely explored in classification, prior studies report that RL can yield robust behavior in control and robotics, supported by worst-case optimization viewpoints that treat environment uncertainty explicitly during learning (Pinto et al., 2017; Rajeswaran et al., 2017; Nilim & El Ghaoui, 2005; Wiesemann et al., 2013; Derman & Mannor, 2020).

For input perturbations more similar to evasion attacks, a lightweight sensitivity penalty discourages large output changes when the input is changed slightly. Prior work demonstrates consistent robustness gains across common algorithms with minimal loss in clean performance, while placing stronger emphasis on early decisions further mitigates behavior drift over time. (Zhang et al., 2020; Yamabe et al., 2024a). Beyond pixel changes, multi-agent studies demonstrate that an opponent can steer a victim policy into harmful behavior without modifying pixels directly, underscoring the need to test opponent-driven threats as well (Gleave et al., 2020; Yamabe et al., 2024b). For safety-critical cases, online selection rules that prefer actions remaining good under bounded input noise have been shown to improve resilience and come with simple certificates (Lütjens et al., 2020). Evidence outside control also points in the same direction: an RL-style generator–classifier training improves robustness to lexical substitutions in text classification (Xu et al., 2019), and RL-based sequential feature acquisition improves resilience when the model must decide which features to read before predicting (Janisch et al., 2020). However, despite these advances in adjacent areas, a systematic comparison where RL serves as the **primary** training paradigm for adversarially robust **image clas-**

**sification**, under standard white-box and black-box attacks and matched training budgets, remains limited.

# 3 PRELIMINARIES

This section introduces the preliminaries necessary for understanding our RL-based training framework. In particular, we focus on concepts that directly support our theoretical view of robustness: policy-gradient objectives, $\epsilon$-greedy exploration, and the notion of input gradients under policy optimization. These analytical tools will be used in Section 7 to support the interpretation of the experimental results reported in Section 6. Our central hypothesis is that models trained by SL and RL differ in their training process: the former relies mainly on gradient descent, while the latter depends on exploration. Concretely, SL optimizes models by directly getting closer to the training dataset, whereas RL optimizes the model's parameters not only by merely training on datasets but also by exploring more sample space. To examine these differences both qualitatively and quantitatively, we use three complementary perspectives (formal definitions and computation details are provided in Appendix A.2): (1) decision-boundary and loss-landscape visualizations, (2) gradient-based indicators in static and dynamic analysis, and (3) predictive uncertainty by entropy:

(1) **Decision-boundary diagrams** visualize classification regions under adversarial perturbations of the input, revealing sharpness / flatness of boundary(Fawzi et al., 2018; Moosavi-Dezfooli et al., 2016). **Loss-landscape diagrams** plot the scalar loss along the attacking direction and perpendicular-attacking direction, making the loss gradient visible (Li et al., 2018; Liu et al., 2020).

(2) **IGV (Input-Gradient Variance)** captures the variance of the input gradient $\nabla_x \mathcal{L}$ under small perturbations of the input (Wang & He, 2021; Agarwal et al., 2022); **dIGV (Directional Input-Gradient Variance)** measures the variability of the direction of the input gradient (Liu et al., 2023; Deng et al., 2023); and **AGN (Average Gradient Norm)** shows how large each update step is (Moosavi-Dezfooli et al., 2016). These static indicators test whether SL and RL training induce different gradient fields even before any attack is applied. To study how gradients evolve during iterative adversarial optimization, the **Gradient Stability Under Attack (GSUA)** diagram is introduced, which records the cosine similarity between consecutive attack gradients. High GSUA indicates a coherent and stable ascent direction; low GSUA signals a noisy or rapidly changing gradient field. We also track the $\ell_2$ **gradient-norm trajectory** across attack steps to separate directional instability from changes in scale. These dynamic measurements reflect how the gradient is changing, which is highly related to adversarial example generation from an adversary.

(3) **Mean predictive entropy** summarizes the dispersion of the model's predictive distribution, indicating how confidently the model distributes the probability mass under clean and perturbed inputs (Smith & Gal, 2018; Kopetzki et al., 2021; Qin et al., 2021; Emde et al., 2024).

# 4 METHODOLOGY

We propose a reinforcement learning-based method for image classification that enhances standard policy gradient optimization with two key components: an Epsilon-Greedy action selection strategy and, optionally, adversarial training via FGSM perturbations. These extensions aim to improve the robustness and generalization of the learned policy beyond what conventional REINFORCE-style algorithms can achieve.

Before describing our reinforcement learning objective, we briefly state the SL baseline used for comparison. Given a training sample $(I, y)$, where $y$ is the ground truth class label and $\hat{p} = f_\theta(I)$ is the predicted probability vector, SL minimizes the standard cross-entropy loss:

$$\mathcal{L}_{CE} = -\sum_{c=1}^{C} \mathbb{I}(y = c) \cdot \log \hat{p}_c, \tag{1}$$

here, $C$ represents the total number of classes, and $\mathbb{I}(y = c)$ is an indicator function that equals 1 if $y = c$ and 0 otherwise. (Additional training details are provided in Appendix A.5).

In standard policy-gradient methods (e.g., REINFORCE), actions $a$ are sampled from a categorical policy $\pi_\theta(a \mid s)$, and gradients follow $\nabla_\theta \log \pi_\theta(a \mid s)$ weighted by returns/advantages; mod-

ern variants such as TRPO/PPO implement stable surrogates (Williams, 1992; Sutton et al., 2000; Schulman et al., 2015; 2017; Mnih et al., 2016; Ahmed et al., 2019; Haarnoja et al., 2018). To ensure adequate exploration in our classification setting, we incorporate an Epsilon-Greedy action selection scheme. Specifically, with probability $\varepsilon_{Greedy}$, the action is sampled uniformly at random, and with probability $1 - \varepsilon_{Greedy}$, it is sampled from the policy's predicted distribution. This simple mechanism introduces explicit stochastic exploration into the learning process and helps the model to avoid premature convergence to suboptimal decision boundaries.

Given an input image $I$, the policy network outputs a vector of class scores, which is then normalized using the softmax function. For numerical stability, we subtract the maximum logit and add a small constant $\epsilon$ before normalization. If the selected action $a_t$ matches the true label $a^*$, a reward of $r_t = 1$ is assigned; otherwise, $r_t = 0$. The loss is then computed as:

$$\mathcal{L}_{\text{PG}} = -\mathbb{E}_{a \sim \pi_\theta} \left[ \log \pi_\theta(a_t|I) \cdot \mathbb{I}(a_t = a^*) \right], \tag{2}$$

and gradients are computed accordingly to update the policy parameters using stochastic gradient ascent. To further improve robustness, particularly under input perturbations or adversarial scenarios, we optionally apply adversarial training using the Fast Gradient Sign Method (FGSM). For a random subset of each batch, we compute the input gradient with respect to the policy loss and apply a perturbation in the direction of the gradient sign, generating adversarial examples $I_{\text{adv}} = \text{clip}(I + \epsilon \cdot \text{sign}(\nabla_I \mathcal{L}_{\text{PG}}), 0, 1)$, where $\epsilon$ represents perturbation magnitude (distinct from the exploration probability $\varepsilon_{\text{Greedy}}$ introduced earlier). These adversarial samples are then combined with the remaining clean samples to form a mixed batch, upon which a second policy gradient update is performed. This adversarial augmentation serves as a regularizer, encouraging the model to learn classification policies that are stable under perturbations and less sensitive to minor variations in input space. In summary, our method introduces exploration via Epsilon-Greedy sampling and enhances robustness through adversarial regularization, providing a more resilient policy learning paradigm for image classification.

## 5 IMPLEMENTATION

### 5.1 BENCHMARKS AND BACKBONES

**Benchmarks:** To systematically evaluate the robustness improvements of RL compared to SL, we conduct experiments on three benchmark datasets: CIFAR-10 (Krizhevsky & Hinton, 2009), CIFAR-100 (Krizhevsky & Hinton, 2009) and ImageNet-100 (Tian et al., 2020), of which the detailed dataset information can be found in Appendix A.3.

**Backbones:** To comprehensively evaluate the robustness of both SL and RL, three neural network models with various complexities are used: a 4-layer CNN, a 6-layer CNN and Resnet18 (He et al., 2016), where the complete model architectures are illustrated in Figure 6 in Appendix A.4.

### 5.2 TRAINING CONFIGURATION

Training configuration consists of two training phases: standard training on clean samples and adversarial training incorporating perturbed examples. For the adversarial training phase, FGSM is used with TRADES (TRadeoff-inspired Adversarial DEfense via Surrogate-loss minimization) (Zhang et al., 2019), where the regularization weight $\beta$ from TRADES tunes the accuracy–robustness trade-off, offering greater flexibility than standard adversarial training. The configuration details are shown in Appendix A.5.

$$\theta^* = \arg\min_\theta \mathbb{E}_{(\mathbf{X},\mathbf{Y})} \left\{ \underbrace{\mathcal{L}(f_\theta(\mathbf{X}), \mathbf{Y})}_{\text{Standard Accuracy}} + \beta \cdot \underbrace{\max_{\mathbf{X}' \in \mathbb{B}(\mathbf{X}, \epsilon)} \mathcal{L}(f_\theta(\mathbf{X}), f_\theta(\mathbf{X}'))}_{\text{Robustness Regularizer}} \right\}, \tag{3}$$

here, $f_\theta$ denotes the model with parameters $\theta$, $\mathcal{L}(\cdot, \cdot)$ represents the cross-entropy loss, $f_\theta(\mathbf{X})$ and $f_\theta(\mathbf{X}')$ are the output of the clean and adversarial examples of the model, respectively. $\beta$ is the regularization parameter, where it is small at early training to prioritize standard accuracy, while it will be large at late training to enhance model robustness.

## 5.3 ATTACK CONFIGURATION

A standard PGD is implemented using the CleverHans library (Papernot et al., 2018), which provides widely validated $\ell_2$-projection and step-size scheduling, ensuring reproducible and comparable results. To further strengthen evaluation, we additionally use AutoAttack (Croce & Hein, 2020b), a standardized ensemble of adaptive and gradient-free attacks (APGD-CE (Croce & Hein, 2020b), APGD-T (Croce & Hein, 2020b), FAB-T (Croce & Hein, 2020a), and SQUARE Andriushchenko et al. (2020)). Both standard PGD and AutoAttack are applied to standard and adversarially trained models. Full attack configurations are provided in Appendix A.7 and Appendix A.11.

# 6 RESULTS AND EMPIRICAL THEORY

## 6.1 MODEL ACCURACY AND ROBUSTNESS

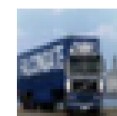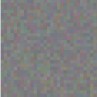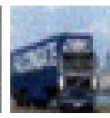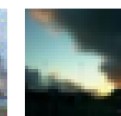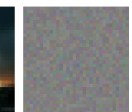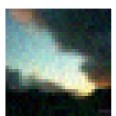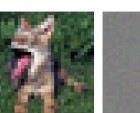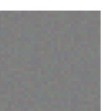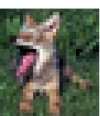

(a) CI-10 ("truck" → "ship")  (b) CI-100 ("cloud" → "turtle")  (c) IN-100 ("coyote" → "terrier")

Figure 2: Visualization of adversarial examples attacked on 6-layer-CNN-SL across benchmark datasets: (a) CIFAR-10 example showing original image (left), additive perturbation (middle), and adversarial image (right); (b) CIFAR-100 example; (c) ImageNet-100 example.

As discussed in Section 5.1, a 4-layer CNN, a 6-layer CNN, and Resnet18 are trained and evaluated on CIFAR-10, CIFAR-100, and ImageNet-100 under adversarial attacks. The main phenomena we highlight are observed consistently across all three backbones. As a representative case, we focus on the 6-layer CNN in the main text, while full results (including 4-layer CNN and ResNet-18) are shown in Appendix A.8. Figure 2 illustrates the adversarial generation process for 6-layer-CNN-SL across all benchmark datasets, showing the original image ($X$), the additive perturbation ($\delta$) and the adversarial image ($X + \delta$). The perturbations remain imperceptible to human observers, but the model misclassifies the image (e.g., CIFAR-10: true label "truck", predicted label before attack "truck", predicted label after attack "ship").

Table 1 shows the model accuracy for the 6-layer-CNN across different datasets. Here, "-SL" denotes SL without adversarial training, "-SL-adv" denotes SL with adversarial training, "-RL" refers to RL without adversarial training, and "-RL-adv" refers to RL with adversarial training. It reveals two key findings: (1) For a clean dataset, the accuracy of SL and SL-adv on CIFAR-10, CIFAR-100, and ImageNet-100 is the highest, and the accuracy of RL and RL-adv is only 3-5% lower. This is expected, since SL directly minimizes the cross-entropy loss with ground-truth labels, providing a strong and stable gradient that favors efficient convergence and higher accuracy. By contrast, RL relies on rewards, which are typically noisier, less aligned with the label distribution, and introduce higher variance, leading to less sample-efficient optimization and thus lower clean accuracy. (2) Although the accuracy of RL and RL-adv is (only) 3-5% lower in a clean dataset, they are the highest in the adversarial datasets compared to the accuracy of SL and SL-adv, which supports our hypothesis that RL provides more model robustness than SL. The further detailed explanation will be discussed in Section 7.

Table 1: Model robustness evaluation (%) across datasets.

| Model | CIFAR-10 | | CIFAR-100 | | ImageNet-100 | |
|---|---|---|---|---|---|---|
| | Clean (%) | AE (%) | Clean (%) | AE (%) | Clean (%) | AE (%) |
| 6-layer-CNN-SL | **90.74*** | 5.00 | **64.75*** | 2.53 | 57.64 | 5.72 |
| 6-layer-CNN-SL-adv | 90.11 | 4.96 | 63.61 | 2.83 | **58.00*** | 5.36 |
| 6-layer-CNN-RL | 88.50 | **55.77*** | 59.80 | 13.06 | 55.60 | 18.04 |
| 6-layer-CNN-RL-adv | 87.63 | 48.63 | 56.54 | **25.51*** | 45.92 | **18.24*** |

Table 2 shows AutoAttack results for 6-layer-CNN models on CIFAR-10, revealing a clear robustness hierarchy: CNN-RL-adv > CNN-SL-adv > CNN-RL > CNN-SL, indicating that RL-adv consistently achieves the strongest adversarial performance across all AutoAttack components, and full results (including CIFAR-100 and ImageNet-100) are shown in Appendix A.11. Compared to standard PGD, non-adversarial RL and SL show similar robustness across all AutoAttack components. This is because AutoAttack leverages adaptive loss objectives, restarts, and gradient-free queries, allowing it to discover adversarial directions even when gradients in RL models are flatter and less stable. Thus, when an adversarial direction is successfully identified, both non-adversarial models are similarly vulnerable. SL-adv achieves higher robustness than these non-adv models due to the enlarged decision-boundary margin introduced by adversarial training. In contrast, RL-adv benefits from two complementary robustness mechanisms: (i) exploration-induced gradient instability, which increases attack optimization difficulty, and (ii) adversarially enforced margins, which ensure decision stability under strong attacks. This dual mechanism enables RL-adv to maintain the highest robustness under AutoAttack, and is consistent with both standard PGD and AutoAttack evaluation.

Table 2: Model robustness evaluation (%) in CIFAR-10 dataset by AutoAttack.

| | Clean (%) | APGD-CE (%) | APGD-T (%) | FAB-T (%) | SQUARE (%) |
|---|---|---|---|---|---|
| 6-layer-CNN-SL | **90.82*** | 15.42 | 13.58 | 13.58 | 11.6 |
| 6-layer-CNN-SL-adv | 87.03 | 24.87 | 21.77 | 21.77 | 19.55 |
| 6-layer-CNN-RL | 88.62 | 16.71 | 13.73 | 13.73 | 11.96 |
| 6-layer-CNN-RL-adv | 86.29 | **36.27*** | **35.4*** | **35.4*** | **32.41*** |

Table 3 shows model accuracy on adversarial examples generated by different source models. Two key observations can be made: (1) For adversarial examples generated from SL, "SL", "SL-adv", and "RL" achieve less than 10% accuracy, whereas for adversarial examples generated from RL, all models maintain above 40% accuracy. This suggests that adversarial examples are substantially easier to generate from SL models, while RL models exhibit a hard-to-generate property, which will be further analyzed in Section 7. (2) Although plain RL demonstrates robustness against adversarial attacks generated from itself (owing to the hard-to-generate property), it remains vulnerable to adversarial examples transferred from weaker models such as SL. In contrast, RL with adversarial training provides robustness against both strong (RL: 54.31% / 46.91%) and weak (SL: 30.88% / 22.56%) adversarial sources, highlighting the necessity of adversarial training when deploying RL-based models.

Table 3: Model robustness evaluation (%) in CIFAR-10 datasets across models.

| | Adversarial Examples (AE) | | | |
|---|---|---|---|---|
| **Model** | **SL (%)** | **SL-adv (%)** | **RL (%)** | **RL-adv (%)** |
| 6-layer-CNN-SL | 5.81 | 7.18 | 48.45 | 43.15 |
| 6-layer-CNN-SL-adv | 9.53 | 5.74 | 50.14 | 44.86 |
| 6-layer-CNN-RL | 8.29 | 7.92 | 48.84 | 42.69 |
| 6-layer-CNN-RL-adv | **30.88*** | **22.56*** | **54.31*** | **46.91*** |

## 6.2 DECISION BOUNDARY AND LOSS GEOMETRY

Before introducing our quantitative indicators, we outline a high-level view of how SL and RL shape the input-loss geometry. The fundamental difference between SL and RL happens in their optimization methods, "Cross Entropy" as formalized by Equation 1 and "Policy Gradient with $\varepsilon_{Greedy}$" as formalized by Equation 2. The cross-entropy loss function provides a deterministic gradient (direction of gradient descent) for updating the model's parameters, accelerating training convergence. However, if this information of deterministic gradients is kept in the training model, it can create stable attack surfaces. Nevertheless, policy gradient with $\epsilon$ employs a loss function with the exploration-exploitation mechanism, where it induces flatter and more unstable gradient directions. This gradient-flattening phenomenon inherently obscures attack pathways and may act as an implicit defense mechanism, whereas the more pronounced gradient structures in SL align with the linear vulnerability hypothesis (Goodfellow et al., 2014).

**Decision boundary:** Figure 3 (a) shows the decision boundary of both SL and RL, where SL's decision boundary should be steeper than RL's decision boundary, because of its deterministic gradients; however, a 2D decision boundary shows limited visual differentiation.

**Loss landscape:** Figure 3 (b) shows that SL has a larger gradient magnitude and a wider dynamic range than the RL on the decision boundary ($(max - min)_{boundary}$ value of SL $>> 10.5$, while $(max - min)_{boundary}$ value of RL $< 10.5$). This loss gradient difference directly impacts adversarial vulnerability: SL's large loss gradients enable efficient perturbation calculation via $\nabla_x \mathcal{L}(x, y)$, whereas RL's small loss gradients inherently resist gradient-based attack optimization. Appendix A.9 provides a first-order theoretical analysis of how gradient magnitude influences adversarial sensitivity.

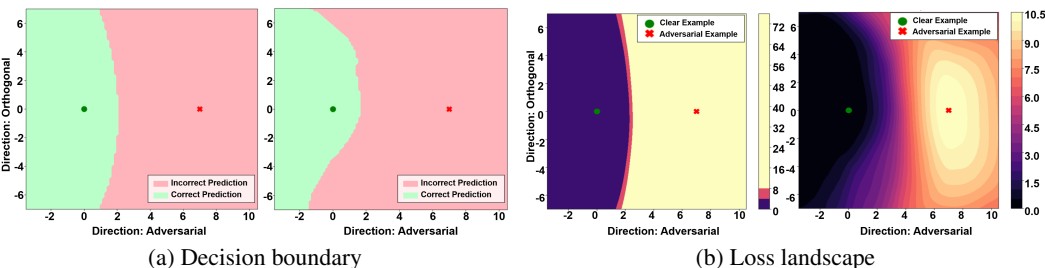

(a) Decision boundary             (b) Loss landscape

Figure 3: Comparative analysis of 6-layer CNN models trained with (left) SL versus (right) RL on one image from CIFAR-10: (a) decision boundary and (b) loss landscape.

### 6.3 GRADIENT INSTABILITY ANALYSIS

To analyze the robustness mechanisms behind SL and RL, three static and two dynamic indicators are evaluated on a 6-layer CNN trained on CIFAR-10. All perturbations are bounded by a conventional adversarial attacking setting ($\epsilon < 8/255$) (MadryLab, 2017).

For the static indicators, the **average gradient norm (AGN)** is larger for SL (2.158) than for RL (1.9527) for the complete CIFAR-10 dataset. This indicates that small input perturbations cause larger loss change in SL, allowing larger effective step size for gradient-based attacks. The **input gradient variance (IGV)** further confirms that SL updates in input space are consistently larger than RL, shown in Figure 4a. In contrast, the **directional input gradient variance (dIGV)** is markedly higher for RL, shown in Figure 4a, reflecting greater instability of gradient directions under perturbations. It is concluded that these results imply that SL is more vulnerable because of its larger and more stable gradients (high AGN/IGV, low dIGV), whereas RL is more robust due to unstable gradient directions (high dIGV) and smaller effective step sizes (low AGN/IGV).

For the dynamic indicators, Figure 4b shows the gradient evolution during PGD attacks. The **gradient stability under attack (GSUA)** between consecutive steps is high for SL (0.8), indicating stable adversarial directions, while RL exhibits low or even negative similarity ($-0.2$), suggesting unstable gradients that hinder attack convergence. Similarly, the $L_2$ **gradient norm** is larger for SL ($\sim 0.6$), allowing faster adversarial progress per iteration, whereas RL's ($\sim 0.1$) smaller gradient norms slow attack optimization.

### 6.4 CALIBRATION-AWARE ROBUSTNESS

Figure 5 shows the mean predictive entropy of SL (CNN-SL), adversarially-trained SL (CNN-SL-adv), and RL (CNN-RL) under varying perturbation magnitudes ($\epsilon$). The RL model consistently produces a higher entropy than both SL and SL-adv.

SL is characterized by stable gradient directions (high cosine similarity, low dIGV) and relatively large gradient norms (high AGN, large $L_2$ norm), allowing gradient-based attack (e.g., PGD) updates to efficiently align with adversarial directions and quickly drive an incorrect logit above the correct one. In contrast, RL has unstable gradient directions (low cosine similarity, high dIGV) and smaller gradient norms (low AGN, small $L_2$ norm), meaning that attack steps tend to fluctuate in

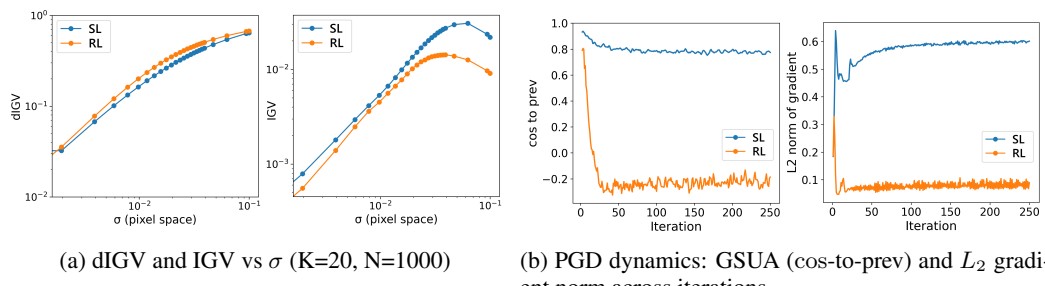

(a) dIGV and IGV vs $\sigma$ (K=20, N=1000)

(b) PGD dynamics: GSUA (cos-to-prev) and $L_2$ gradient norm across iterations

Figure 4: Comparison of 6-layer CNNs trained with SL (blue) and RL (orange) on CIFAR-10: (a) Gradient variance indicators (dIGV, IGV) as a function of input noise scale $\sigma$ and (b) PGD attack dynamics showing cosine similarity to the previous step and gradient $L_2$ norm across iterations.

direction and have smaller effective magnitudes. Even when attack steps move toward an adversarial direction, the increase of the incorrect logit relative to the correct one is much slower due to small gradient norms. As a result, SL tends to yield highly confident but incorrect predictions (e.g., [0.01, 0.01, 0.98], low entropy), as shown in Figure 5 (right), whereas RL outputs remain less sharply peaked (e.g., [0.3, 0.3, 0.4], higher entropy). From the perspective of calibration-aware robustness, these findings suggest that SL models are more prone to overconfident errors, while RL models maintain higher predictive uncertainty even when misclassifying.

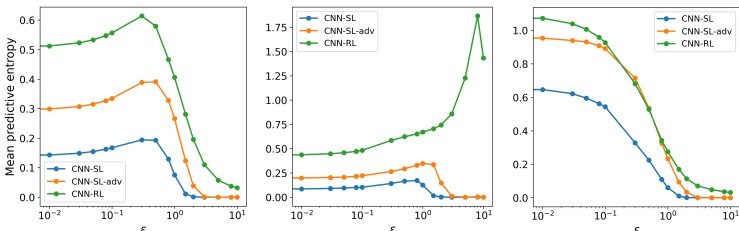

Figure 5: Predictive entropy vs. $\epsilon$ (perturbation magnitude) for 6-layer CNN Architecture (CNN-SL (blue), CNN-SL-adv (orange), CNN-RL (green)) on CIFAR-10: total (left), correct prediction (middle), wrong prediction (right).

# 7 UNIFIED THEORETICAL ANALYSIS OF ROBUSTNESS MECHANISMS

The empirical results across multiple benchmarks provide converging evidence that reinforcement learning (RL) models exhibit stronger robustness than SL in Section 6.1. As shown in Table 3, adversarial accuracy remains consistently higher for SL, SL-adv, RL and RL-adv on the adversarial examples (AEs) generated from RL and RL-adv. This indicates that AEs are inherently harder to exploit from RL-trained models. We emphasize that this section provides a unified, hypothesis-driven interpretation of the robustness phenomena observed in our experiments, rather than a fully general mathematical guarantee that holds for all possible architectures and training regimes. We now discuss why this phenomenon arises by linking back to the preceding analyses.

At the core lies the distinction between the training process. In SL, each labeled sample directly defines the loss that yields a clear, low-variance gradient for updating parameters. By contrast, RL introduces an exploration–exploitation step between sampled data and parameter updates. This exploration acts as a form of implicit regularization, increasing the variance of the gradient signal. Section 6.2 shows that SL has a large and clear loss gradient, whereas RL has a lower loss gradient near the decision boundary.

This gradient information introduces adversarial vulnerabilities. From our gradient instability analysis (Section 6.3), SL consistently demonstrated larger gradient magnitudes (high AGN/IGV) and more stable directions (low dIGV, high cosine similarity). Consequently, adversaries can reliably

recover a descent direction, and each PGD step makes efficient progress due to the large gradient norm. RL, on the other hand, exhibits unstable gradient directions (high dIGV, low or even negative cosine similarity) and smaller norms, which jointly slow down adversarial optimization. This explains why RL-trained models are more robust than SL-trained models.

The implications are also evident in the calibration-aware robustness analysis (Section 6.4). Under perturbations, SL models tend to produce highly confident but incorrect predictions (low entropy), reflecting the fact that attacks can push the logits of an incorrect class decisively above the true class. RL models, however, maintain higher entropy under misclassification, suggesting that their predictions will always retain uncertainty. This aligns with the gradient-based explanation: because RL's adversarial directions are less recoverable, the induced misclassifications remain and the entropy of results is higher.

Taken together, these findings highlight a distinct robustness mechanism in RL: adversarial attacks are hampered by gradient instability and reduced gradient magnitudes, both of which stem from the exploration inherent in the training process. Nevertheless, our results also show that RL alone is not sufficient for robust deployment. As indicated in Table 3, performing an adversarial attack is difficult when generating AEs from RL; however, it can be successful when using AEs from other weak models. Thus, while RL provides a promising foundation by obscuring gradient-based attack pathways, adversarial training remains necessary to harden RL models against stronger or adaptive adversaries.

## 8 CONCLUSION

We introduced a reinforcement learning framework that integrates $\varepsilon$-greedy exploration with adversarial training to improve the robustness of neural networks in image classification. Supported by extensive experiments across datasets and models, our study demonstrates that RL-based training achieves stronger robustness than SL, both quantitatively and qualitatively. These findings not only highlight that reinforcement learning can be an effective defense strategy, but also provide a stepping stone towards understanding robustness mechanisms in deep learning, with potential implications beyond image classification tasks.

## 9 LIMITATION AND FUTURE WORK

Although our study provides a systematic understanding of why reinforcement learning improves adversarial robustness, several limitations remain. Our robustness analysis focuses primarily on convolutional architectures; extending RL-based robustness methods to Transformer or hybrid models requires a separate experimental setup and is left for future work. In addition, RL training is substantially less efficient than supervised learning due to its exploration burden, and improving this efficiency is essential for practical deployment.

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

# A   APPENDIX

## A.1   TERMINOLOGY AND NOTATION

### A.1.1   GENERAL NOTATION

- $I \in [0,1]^{H \times W \times C}$: an input image of size height $H$, width $W$, with color channels $C$.
- $y \in \{1, \ldots, C_{\mathrm{cls}}\}$: the true class label; and $\mathbf{e}_y$ indicates its one-hot vector.
- $f_\theta(I)$: neural network mapping an image to logits $z_\theta(I) \in \mathbb{R}^{C_{\mathrm{cls}}}$.
- $\pi_\theta(a \mid I) = \mathrm{softmax}(z_\theta(I))$: the categorical distribution over classes (policy).
- $\max/\min$, $\mathbb{E}[\cdot]$, $\mathbb{P}[\cdot]$: maximum/minimum, expectation, probability.
- $B_p(\varepsilon) = \{\, \delta \,:\, \|\delta\|_p \leq \varepsilon \,\}$: closed $\ell_p$-ball of radius $\varepsilon$.
- $\mathrm{Proj}_{B_p(\varepsilon)}(\cdot)$: projection onto $\ell_p$-ball (clipping to the budget).
- $\mathrm{Uniform}(a)$: uniform distribution over the discrete action/class space.
- $\log$: natural logarithm.
- $\mathrm{KL}(p\|q)$: the Kullback–Leibler divergence.
- Numerical stability constant $\epsilon_{\mathrm{num}}$: a tiny constant (e.g., $10^{-8}$) used only inside $\log(\cdot)$ to avoid $\log 0$.

### A.1.2   RL FORMULATION FOR CLASSIFICATION

- **Single-step Markov Decision Process (MDP):** We formulate classification as a one-step decision process in which the state is the image $I$, the action $a$ denotes the predicted class; and the reward is given by $r(I, a) = \mathbb{1}[a = y]$.
- **Policy $\pi_\theta$**: A categorical distribution over classes parameterized by network logits $z_\theta(I)$.
- **Behavior policy vs. target policy**: The behavior policy $\tilde{\pi}$ is used to generate actions during learning, while the target policy $\pi$ demotes the optimized policy. If $\tilde{\pi} \neq \pi$, unbiased policy-gradient updates require importance weighting.
- **Epsilon-Greedy ($\varepsilon_{\mathbf{greedy}}$)**: A mixture policy

$$\tilde{\pi}_\theta(a \mid I) = (1 - \varepsilon_{\mathrm{greedy}})\, \pi_\theta(a \mid I) + \varepsilon_{\mathrm{greedy}}\, \mathrm{Uniform}(a),$$

with $\varepsilon_{\mathrm{greedy}} \in [0,1]$ often decayed over training to balance exploration and exploitation. Note that $\varepsilon_{\mathrm{greedy}}$ is distinct from the adversarial budget $\varepsilon_{\mathrm{adv}}$.
- **REINFORCE / Policy Gradient**: For sampled an action $a \sim \tilde{\pi}$, the policy-gradient estimator is

$$\nabla_\theta \mathcal{L}_{\mathrm{PG}} = -\, \mathbb{E}\big[ w(a)\, (r - b)\, \nabla_\theta \log \pi_\theta(a \mid I) \big],$$

where $w(a) = \pi_\theta(a \mid I)/\tilde{\pi}_\theta(a \mid I)$ is the importance weight (often set to 1 in practice), and $b$ is a baseline to reduce variance.
- **TRPO/PPO**: Trust Region Policy Optimization (TRPO) and Proximal Policy Optimization (PPO) are modern policy gradient methods that implement stable parameter updates through constrained optimization, avoiding the large policy changes that can destabilize training in vanilla policy gradient methods.
- **Baseline $b$**: A baseline is a control variate (e.g., moving average of rewards or a learned value) that reduces gradient variance without biasing the estimate.
- **Entropy regularization**: An entropy regularization is an auxiliary term $-\lambda\, \mathcal{H}(\pi_\theta(\cdot \mid I))$ that encourages exploration and prevents overly peaked policies.
- **Advantage**: $A(I, a) = r(I, a) - b$, represents how much an action is better than the baseline.
- **Logit / Softmax**: $z_\theta(I)$ are pre-softmax scores and probabilities are $\pi_\theta = \mathrm{softmax}(z_\theta)$. For stability, $\max_k z_k$ is subtracted before applying softmax and a small $\epsilon_{\mathrm{num}}$ is added inside the $\log$.

### A.1.3 ADVERSARIAL ROBUSTNESS

- **Adversarial budget** $\varepsilon_{\mathbf{adv}}$: The radius of allowed perturbations in the chosen norm $\ell_p$ (typically $\ell_\infty$ or $\ell_2$).

- **Robust risk**: $\mathcal{R}_{\mathrm{rob}}(\theta) = \mathbb{E}_{(x,y)}\big[\max_{\delta \in B_p(\varepsilon_{\mathrm{adv}})} \mathcal{L}(f_\theta(x+\delta), y)\big]$, where $\mathcal{L}$ is typically cross-entropy or a policy loss.

- **Robust accuracy**: In practice, robust accuracy is approximated by accuracy under a strong attack (white-box, multi-step, possibly multi-restart) and is denoted by $\mathrm{Acc}_{\mathrm{rob}} = \mathbb{P}[f_\theta(x + \delta) = y, \ \forall \delta \in B_p(\varepsilon_{\mathrm{adv}})]$.

- **White-box / Black-box / Transfer**: In white-box, the parameters and gradients are known; in black-box, there is only query access; and transfer attacks use adversarial examples generated on a surrogate model.

- **Random start**: Attacks (e.g., PGD) initialize within $B_p(\varepsilon_{\mathrm{adv}})$ to avoid deterministic local traps.

- **Label leaking**: An artifact where adversarial training with single-step gradients can leak label information, which can be mitigated by random starts, multi-step attacks, or TRADES-style regularization.

- **Obfuscated gradients / Gradient masking**: Using these methods, models appear robust because gradients are uninformative or broken; sanity checks (below) must rule this out. Proper sanity checks are necessary to verify true robustness.

### A.1.4 ATTACKS AND INNER MAXIMIZATION

- **FGSM**: Fast Gradient Sign Method, where the adversarial example is computed as

$$x_{\mathrm{adv}} = \mathrm{clip}_{[0,1]}\big(x + \varepsilon_{\mathrm{adv}} \cdot \mathrm{sign}(\nabla_x \mathcal{L})\big).$$

- **PGD-$k$**: PGD with $k$ steps and step size $\alpha$, where the iterative update using:

$$x^{t+1} = \mathrm{Proj}_{B_p(\varepsilon_{\mathrm{adv}})}\big(x^t + \alpha \cdot \mathrm{sign}(\nabla_x \mathcal{L}(x^t))\big).$$

Variants include momentum (MI-FGSM), BIM (iterative FGSM), and restarts $R$.

- **AutoAttack (AA)**: A standardized, strong parameter-free ensemble (e.g., APGD-CE, APGD-T, FAB-T, and a black box attack known as Square) Croce & Hein (2020b). AA is widely used to benchmark true robustness.

- **APGD-CE**: The Auto-PGD variant optimizing the cross-entropy loss (untargeted) Croce & Hein (2020b).

- **APGD-T**: The Auto-PGD variant in a targeted setting (often optimizing a margin-based or DLR loss) Croce & Hein (2020b).

- **FAB-T**: The targeted version of the FAB Attack (Fast Adaptive Boundary) attack that minimizes perturbation norm under a target misclassification constraint Croce & Hein (2020a).

- **SQUARE**: The Square Attack: a query-efficient black-box adversarial attack based on random search of square-shaped perturbations Andriushchenko et al. (2020).

- **Clarin Winger (CW) attack** Carlini & Wagner (2017): Optimization-based attack that minimizes a margin-based objective under norm constraints.

- **FAB / Square**: FAB is a decision-based strong attack; Square is a black-box, score-based attack using square-shaped perturbations.

- **DLR loss**: Difference of Logits Ratio loss used in APGD-DLR to avoid saturation of cross-entropy under strong perturbations.

- **Step size** $\alpha$: Gradient step magnitude within PGD; tuned relative to $\varepsilon_{\mathrm{adv}}$ (e.g., $\alpha = \frac{2}{255}$ under $\ell_\infty$).

- **Restarts** $R$: Number of random re-initializations for multi-start attacks; larger $R$ increases attack strength.

### A.1.5 ROBUST TRAINING OBJECTIVES

- **Standard (ERM) training**: Minimizes $\mathbb{E}[\mathcal{L}(f_\theta(x), y)]$ on clean data; high clean accuracy but vulnerable to adversarial perturbations.

- **Adversarial training**: Minimizes expected $\max_{\delta \in B_p(\varepsilon_{\text{adv}})} \mathcal{L}(f_\theta(x + \delta), y)$ by alternating inner maximization (attack) and parameter updates.

- **TRADES**: Balances natural accuracy and robustness by solving

$$\min_\theta \mathbb{E}\Big[\underbrace{\text{CE}(f_\theta(x), y)}_{\text{natural}} + \beta \cdot \underbrace{\text{KL}\big(\pi_\theta(\cdot \mid x) \,\|\, \pi_\theta(\cdot \mid x_{\text{adv}})\big)}_{\text{robust}}\Big],$$

  where $x_{\text{adv}}$ is found by maximizing the KL term under $B_p(\varepsilon_{\text{adv}})$. $\beta > 0$ trades off accuracy and robustness.

- **Consistency regularization (logit/policy matching)**: Encourages predictions on $x$ and $x_{\text{adv}}$ to be close (e.g., via KL), stabilizing decision boundaries.

- **Label smoothing**: Replaces one-hot target with a softened distribution to reduce overconfidence and improve calibration.

### A.1.6 GEOMETRY, GRADIENTS, AND DIAGNOSTICS

- **Loss landscape flatness**: Informally, a flatter neighborhood around inputs or parameters indicates smaller gradients and less exploitable directions; proxies include gradient norm, local Lipschitz, or Hessian trace.

- **Gradient norm**: Magnitude $\|\nabla_x \mathcal{L}\|_p$; smaller norms often correlate with higher resistance to small-norm attacks (not sufficient alone).

- **GSUA (Gradient Similarity / Sign Uniformity Analysis)**: Measures gradient *directional stability* across attack iterations. A typical definition at iteration $t$ uses cosine similarity

$$\text{GSUA}_t = \cos\big(\nabla_x \mathcal{L}(x^t), \, \nabla_x \mathcal{L}(x^{t-1})\big),$$

  averaged over $t$ and samples. Lower (or negative) GSUA indicates rapidly changing attack directions, making optimization harder; high GSUA implies stable, aligned directions that favor attacker success. A sign-based variant reports the fraction of coordinates with matching gradient signs.

- **Margin**: Logit margin $m = z_y - \max_{k \neq y} z_k$; larger margins generally imply higher confidence and sometimes better robustness.

- **Confidence / Predictive entropy**: Confidence $= \max_k \pi_\theta(k \mid x)$; entropy $\mathcal{H}(\pi_\theta(\cdot \mid x))$ quantifies uncertainty. Robust models tend to avoid extreme confidence on perturbed inputs.

- **Sanity checks against gradient masking**: Robust accuracy should *decrease* (not increase) with more attack steps, larger $\varepsilon_{\text{adv}}$, or stronger restarts; black-box and transfer attacks should not outperform white-box; gradient-free attacks (e.g., SPSA) should not catastrophically outperform white-box baselines.

### A.1.7 EVALUATION PROTOCOLS AND METRICS

- **Clean accuracy**: Top-1 accuracy on unperturbed data.

- **Adversarial accuracy (robust accuracy)**: Top-1 accuracy under a specified attack (norm, $\varepsilon_{\text{adv}}$, steps, restarts).

- **Transfer robustness**: Accuracy on adversarial examples generated from different source models.

- **ECE (Expected Calibration Error)**: With $M$ confidence bins, ECE $= \sum_{m=1}^{M} \frac{|B_m|}{n} \big|\text{acc}(B_m) - \text{conf}(B_m)\big|$; measures prediction calibration.

- **Brier score / NLL**: Calibration-related metrics; lower is better.

- **Top-1 / Top-5**: Standard accuracy metrics for multi-class evaluation.

- **Report essentials**: Always specify norm ($\ell_\infty, \ell_2$), $\varepsilon_{\text{adv}}$, step size $\alpha$, iterations $k$, restarts $R$, random start, and attack variants (e.g., APGD-CE/DLR).

### A.1.8 TRAINING DETAILS AND REGULARIZATION

- **Learning rate schedule**: Cosine/step/linear decay; should be reported with warmup if any.
- **Gradient clipping**: Bounds on parameter gradients (e.g., global norm clipping) to stabilize training.
- **Data normalization**: Per-channel mean/std normalization; report exact constants.
- **Data augmentation**: Random crop/flip/color jitter, CutMix/Mixup, etc.; can interact with robustness.
- **Temperature scaling**: Post-hoc calibration by dividing logits by $T > 0$ before softmax.

### A.1.9 DISAMBIGUATION OF EPSILONS

- $\varepsilon_{\text{greedy}}$: *Exploration rate* in $\varepsilon$-greedy behavior policy (probability of sampling a random action).
- $\varepsilon_{\text{adv}}$: *Adversarial perturbation budget* (radius of the $\ell_p$-ball used by the attacker).
- $\epsilon_{\text{num}}$: Tiny numeric constant inside $\log$ for stability (e.g., $10^{-8}$); never to be confused with the above.

### A.1.10 COMMON PITFALLS (PRACTICAL NOTES)

- **Softmax stability**: Subtract $\max_k z_k$ before softmax; add $\epsilon_{\text{num}}$ only when taking logs.
- **Behavior vs. target mismatch**: If sampling from $\tilde{\pi}$ but optimizing $\pi$, document whether importance weighting is used ($w = \pi/\tilde{\pi}$); otherwise clarify the estimator is biased but lower variance.
- **Underpowered inner maximization**: Too few PGD steps, lack of random start, or small $\alpha$ can overestimate robustness; report full attack specs.
- **Overconfidence on perturbed data**: Check entropy/confidence and ECE under attacks to avoid brittle decision boundaries.

## A.2 ROBUSTNESS INDICATORS

**Decision boundary diagrams** illustrates the classification regions under adversarial attack, and **loss landscape diagrams** visualizes the loss gradient information. Both diagrams are drawn under two gradient directions: standard adversarial attack gradient and orthogonal-direction attacks gradient to draw a 2D diagram, where perturbations are constrained to be perpendicular to the gradient ascent direction ($\nabla_x \mathcal{L} \perp \delta$).

**IGV (Input Gradient Variance)** calculates value of gradient according to the input variance:

$$IGV = \mathbb{E}_{x \sim \mathcal{D}} \left\{ \mathbb{E}_{\epsilon \sim \mathcal{N}(0,\sigma^2)} \left[ Var(\nabla_x \mathcal{L}(x + \epsilon, \hat{y})) \right] \right\}, \tag{4}$$

where $\epsilon$ is the Gaussian noise added to the input sample, $x$ is the input sample, $\hat{y}$ is the predicted sample, $Var(\cdot)$ is the gradient variance, $\nabla_x \mathcal{L}(\cdot)$ is the gradient.

**dIGV (direction Input Gradient Variance)** calculates the direction of gradient according to the input variance:

$$\text{dIGV} = \mathbb{E}_{x \sim \mathcal{D}} \left\{ \mathbb{E}_{\epsilon \sim \mathcal{N}(0,\sigma^2)} \left[ 1 - \left\langle \frac{g^{(\epsilon)}}{\|g^{(\epsilon)}\|_2}, \bar{u} \right\rangle \right] \right\},$$

$$\bar{u} = \frac{\mathbb{E}_\epsilon \left( \frac{g^{(\epsilon)}}{\|g^{(\epsilon)}\|_2} \right)}{\left\| \mathbb{E}_\epsilon \left( \frac{g^{(\epsilon)}}{\|g^{(\epsilon)}\|_2} \right) \right\|_2}, \tag{5}$$

$$g^{(\epsilon)} = \nabla_x \mathcal{L}(x + \epsilon, \hat{y}),$$

where $g$ is the attack gradient.

**AGN (Average Gradient Norm)** calculates the sensitivity of gradient according to the input variance:

$$AGN = \mathbb{E}_{x \sim \mathcal{D}}[||(\nabla_x \mathcal{L}(x, y)||_2], \tag{6}$$

where $y$ is the true label, $\mathcal{L}(\cdot)$ is the loss function.

**Gradient stability under attack (GSUA) diagram** is drawn to visualize the gradient stability under attack. It is calculated by calculating the cosine (similarity) between two steps of attack gradient:

$$\text{GSUA}^{(t)} = \cos\theta^{(t)} = \frac{g^{(t)} \cdot g^{(t-1)}}{\|g^{(t)}\|\|g^{(t-1)}\|}, g^{(t)} = \nabla_x \mathcal{L}(x^{(t)}, y), \tag{7}$$

where $t$ is the step.

**Mean predictive entropy** $H$ is used to represent the dispersibility of predicted output:

$$H = \frac{1}{N}\sum_{i=1}^{N}\left[-\sum_{k=1}^{C} p_k(\mathbf{x}_i) \cdot \log p_k(\mathbf{x}_i)\right], \tag{8}$$

where $N$ is the number of test samples, $C$ is the number of classes, $p_k(\mathbf{x}_i)$ denotes the predicted probability of class $k$ on input $x_i$ (either clean or adversarial).

### A.3 DATASETS

Table 4: Benchmark dataset specifications.

| Dataset | Training/Test samples | Image Size | Classes |
|---------|----------------------|------------|---------|
| CIFAR-10 | 50,000/10,000 | 32×32×3 | 10 |
| CIFAR-100 | 50,000/10,000 | 32×32×3 | 100 |
| ImageNet-100 | 126,689/5,000 | 224×224×3 | 100 |

### A.4 MODEL ARCHITECTURES

### A.5 SUPERVISED LEARNING

In conventional image classification tasks, supervised learning is a widely adopted approach (Krizhevsky et al., 2012; He et al., 2016; Dosovitskiy et al., 2020). The fundamental objective is to train a neural network model $f_\theta(I)$ to map an input image $I$ to a probability distribution over predefined classes. The model parameters $\theta$ are optimized by minimizing the discrepancy between predicted outputs and ground truth labels using the cross-entropy loss function. Given a training sample $(I, y)$, where $y$ is the ground truth class label and $\hat{p} = f_\theta(I)$ is the predicted probability vector, the cross-entropy loss is defined as:

$$\mathcal{L}_{CE} = -\sum_{c=1}^{C} \mathbb{I}(y = c) \cdot \log \hat{p}_c. \tag{9}$$

Here, $C$ represents the total number of classes, and $\mathbb{I}(y = c)$ is an indicator function that equals 1 if $y = c$ and 0 otherwise. The overall optimization objective is to minimize the expected cross-entropy loss across the training dataset:

$$\theta^* = \arg\min_\theta \mathbb{E}_{(I,y)\sim D}\left[\mathcal{L}_{CE}(f_\theta(I), y)\right]. \tag{10}$$

Training is performed via backpropagation and gradient-based optimization methods, allowing the model to progressively learn discriminative features for accurate classification. This supervised

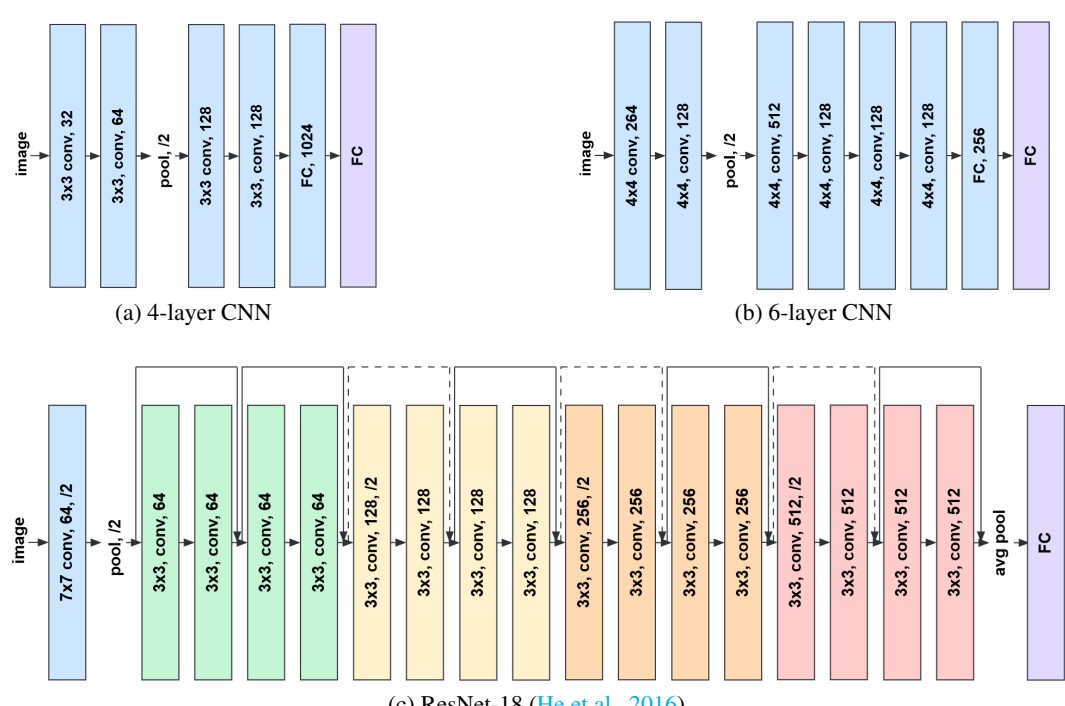

(a) 4-layer CNN          (b) 6-layer CNN

(c) ResNet-18 (He et al., 2016)

Figure 6: Model architectures: (a) 4-layer CNN, (b) 6-layer CNN, and (c) ResNet-18 with residual connections.

approach, with its well-defined loss function, ensures stable convergence and provides a reliable baseline for image classification tasks.

For implementation, the training configuration implements three key mechanisms: (1) a regularization parameter $\beta$ adjustment (initial $\beta = 1$ for accuracy focus, progressing to $\beta = 6$ for robustness); (2) cyclic learning rate scheduling between 0.1 and 0.001; and (3) gradient clipping with threshold 1.0. Each model is trained until convergence on each target dataset (CIFAR-10/CIFAR-100/ImageNet-100), with early stopping based on validation performance.

### A.6 REINFORCMENT LEARNING

Reinforcement learning (RL) is a framework that enables agents to learn optimal decision-making strategies through interactions with an environment, guided by a system of rewards and penalties. Unlike SL, which relies on labeled datasets, RL focuses on learning policies that maximize cumulative rewards over time. The primary objective of RL is to maximize the expected return, which is commonly approximated using the Bellman equation:

$$Q(s, a) = \mathbb{E}\left[r_t + \gamma \max_{a'} Q(s_{t+1}, a')\right]. \tag{11}$$

In this equation, $Q(s, a)$ represents the action-value function, estimating the expected return when taking action $a$ in state $s$. The term $\max_{a'} Q(s_{t+1}, a')$ denotes the maximum expected future reward from the subsequent state $s_{t+1}$, encapsulating the agent's objective to select actions that maximize long-term rewards.

For the task of image classification, we conceptualize the problem as an RL scenario where each classification decision is treated as an action performed by the agent. Specifically, for each input image $I$, the agent selects a class $a$ from a predefined set of possible classes. The reward structure is defined as follows:

- **Correct Classification**: If the agent correctly classifies the image, it receives a reward $r = 1$.

- **Incorrect Classification**: If the agent misclassifies the image, it receives a reward $r = 0$.

This binary reward mechanism simplifies the optimization objective, directing the agent's learning process towards maximizing the number of correct classifications.

The objective is to maximize the expected return $J(\theta)$, which, in this context, corresponds to increasing the number of correctly classified images. Formally, the objective can be expressed as:

$$J(\theta) = \mathbb{E}_{\pi_\theta} \left[ \sum_{t=1}^{T} r_t \right]. \tag{12}$$

Given the binary nature of the rewards, this objective simplifies to maximizing the expected number of correct classifications across the dataset.

To optimize the policy, we employ the REINFORCE algorithm, a fundamental policy gradient method. The gradient of the objective function with respect to the policy parameters $\theta$ is given by:

$$\nabla_\theta J(\theta) = \mathbb{E}_{\pi_\theta} \left[ \nabla_\theta \log \pi_\theta(a_t|s_t) \cdot r_t \right]. \tag{13}$$

Substituting the defined reward structure, the gradient can be rewritten as:

$$\nabla_\theta J(\theta) = \mathbb{E}_{\pi_\theta} \left[ \nabla_\theta \log \pi_\theta(a_t|s_t) \cdot \mathbb{I}(a_t = a^*) \right]. \tag{14}$$

Here, $\mathbb{I}(a_t = a^*)$ is an indicator function that equals 1 if the action $a_t$ corresponds to the correct class $a^*$, and 0 otherwise. This formulation ensures that the policy is updated to increase the probability of actions leading to correct classifications. Specifically, when an image is correctly classified, the gradient update reinforces the chosen action by enhancing its probability, thereby making the policy more likely to select the correct class in future similar instances. Conversely, incorrect classifications do not contribute to the gradient, as their associated reward is zero.

In practice, the policy $\pi_\theta(a|s)$ is parameterized using a neural network, where the input is the image $I$, and the output is a probability distribution over the possible classes. The network parameters $\theta$ are updated using stochastic gradient ascent based on the policy gradient estimate derived from Equation equation 14.

The training procedure involves the following steps:

1. **Forward Pass**: For each image $I$ in the training set, compute the action probabilities $\pi_\theta(a|I)$ using the current policy network.

2. **Action Selection**: Sample an action $a$ (i.e., predict a class) based on the computed probabilities.

3. **Reward Assignment**: Assign a reward $r$ based on whether the predicted class $a$ matches the true class $a^*$.

4. **Gradient Update**: Compute the gradient $\nabla_\theta \log \pi_\theta(a|I) \cdot r$ and update the policy parameters $\theta$ using gradient ascent.

This approach enables the model to learn a policy that maximizes the expected number of correct classifications. By focusing on actions that lead to accurate predictions, the model potentially enhances its robustness against adversarial attacks, as it reinforces strategies that yield reliable classification outcomes.

A.7 ADVERSARIAL ATTACK: STANDARD PROJECTED GRADIENT DESCENT

Adversarial attacks are a class of security threats in machine learning where an adversary can design imperceptible perturbations to input data (e.g. images) to deceive a trained model into making

incorrect predictions, where it was first systematically studied by Szegedy et al. (2014). Among the various adversarial attack methods, the Fast Gradient Sign Method (FGSM) is a fundamental attack introduced by Goodfellow et al. (2014), where the adversarial example $x'$ is generated as:

$$x' = x + \epsilon sign(\nabla_x J(\theta, x, y)). \tag{15}$$

Let $\theta$ be the parameters of a model, $x$ the input to the model, $y$ the targets associated with $x$, $\epsilon$ the perturbation budget, and $J(\theta, x, y)$ be the model's loss function, and $\nabla_x J(\theta, x, y)$ be the gradient of the loss with respect to the input. FGSM efficiently computes gradient signs of the model's loss function with respect to the input to create bounded perturbations with a small perturbation budget $\epsilon$, serving as both an effective attack and baseline for more advanced methods.

Projected Gradient Descent (PGD), introduced by Madry et al. (2018), represents a more advanced adversarial attack by extending the single-step FGSM into an iterative optimization framework with projection constraints, where it generates stronger adversarial examples through:

$$x^{t+1} = \Pi_{x+\mathcal{S}} \left( x^t + \alpha \cdot \text{sign}(\nabla_x J(\theta, x^t, y)) \right). \tag{16}$$

Here, $x^t$ is the adversarial example at iteration $t$, $\alpha$ is the step size (typically $\alpha = \epsilon/T$ for $T$ iterations), $\Pi_{x+S}$ projects the perturbation on the allowed perturbation space around $x$, $\mathcal{S} (= \delta ||\delta|_\infty \leq \epsilon)$ defines the $\ell_\infty$-bounded perturbation space, where in practical $\Pi_{x+\mathcal{S}}(x') = clip(x', x - \epsilon, x + \epsilon)$, $\epsilon$ the perturbation budget.

For our experiments across CIFAR-10, CIFAR-100, and ImageNet-100 datasets, we establish maximum perturbation bounds of $\epsilon$ = 7, 7, and 3.5, respectively. These values align with conventional adversarial training benchmarks (MadryLab, 2017; Madry et al., 2018), where typical perturbation magnitudes remain below $\epsilon$ = 8.0 on the 0-255 pixel intensity scale.

### A.8 COMPLETE MODEL PERFORMANCE

We evaluate the robustness of 4-layer CNN, 6-layer CNN, ResNet-18 on CIFAR-10, CIFAR-100, and ImageNet-100 using non-targeted $\ell_2$ PGD ($K = 250$, step size $\alpha = 0.3$, $\epsilon = 7$). Unless otherwise noted, all results are performed on the test set. Perturbations are applied in the input space before normalization, where we attack the pre-normalized images and then apply dataset normalization. Adversarial examples are clipped to the valid image range.

Additionally, we also considered DenseNet-121 model and the Places365 dataset. Due to the computational cost of reinforcement-learning-based training on large models and datasets, only incomplete evaluations are shown for these settings and leave full RL-based evaluation to future work.

#### A.8.1 CIFAR-10

The complete performance on CIFAR-10 across 4-layer CNN, 6-layer CNN, ResNet-18, ResNet-50, and DenseNet-121 is shown in Table 5. Transfer analyses for these architectures are summarized in Table 6, Table 7, Table 8 and Table 9, respectively.

#### A.8.2 CIFAR-100

The complete performance on CIFAR-100 across 4-layer CNN, 6-layer CNN, ResNet-18, and DenseNet-121 is shown in Table 10. Transfer analyses for these architectures are summarized in Table 11, Table 12, Table 13 and Table 14, respectively.

#### A.8.3 IMAGENET-100

The complete performance on ImageNet-100 across a 4-layer CNN, 6-layer CNN, and ResNet-18 is shown in Table 15. Since our main analyses focus on CIFAR-10/100, we treat ImageNet-100 as a supplementary scale check for verification; therefore, the ImageNet-100 results do not contain transfer analysis.

Table 5: Model robustness evaluation (train, test, and AEs) in CIFAR-10 datasets across models, evaluation under PGD-250, $\ell_2$, $\epsilon = 7$, step size $\alpha = 0.3$, non-targeted.

| Model | Clean train (%) | Clean test (%) | AE (%) |
|---|---|---|---|
| (SL) 4-layer-CNN | 86.53 | 82.03 | 5.90 |
| (SL) 4-layer-CNN-adv | 84.98 | 81.25 | 7.01 |
| (RL) 4-layer-CNN | 88.85 | 83.07 | 36.66 |
| (RL) 4-layer-CNN-adv | 88.57 | 82.94 | 35.67 |
| (SL) 6-layer-CNN | 98.30 | 90.74 | 5.00 |
| (SL) 6-layer-CNN-adv | 96.81 | 90.11 | 4.96 |
| (RL) 6-layer-CNN | 95.93 | 88.50 | 55.77 |
| (RL) 6-layer-CNN-adv | 94.12 | 87.63 | 48.63 |
| (SL) Resnet18 | 99.91 | 91.56 | 46.08 |
| (SL) Resnet18-pt | 99.99 | 95.94 | 67.61 |
| (SL) Resnet18-adv | 99.35 | 90.84 | 28.47 |
| (SL) Resnet18-pt-adv | 99.97 | 95.56 | 56.96 |
| (RL) Resnet18 | 98.03 | 91.40 | 75.44 |
| (RL) Resnet18-pt | 98.21 | 94.22 | 65.07 |
| (RL) Resnet18-adv | 97.15 | 90.40 | 70.09 |
| (RL) Resnet18-pt-adv | 97.38 | 93.73 | 68.40 |
| (SL) Resnet50 | 99.97 | 91.85 | 58.19 |
| (SL) Resnet50-adv | 99.23 | 90.90 | 42.86 |
| (RL) Resnet50 | 97.64 | 90.92 | 72.48 |
| (RL) Resnet50-adv | 96.94 | 90.66 | 68.58 |
| (SL) Densenet121 | 99.90 | 90.09 | 31.33 |
| (SL) Densenet121-pt | 99.996 | 96.97 | 52.00 |
| (SL) Densenet121-adv | 99.05 | 89.72 | 15.77 |
| (SL) Densenet121-pt-adv | 99.98 | 96.74 | 28.80 |
| (RL) Densenet121 | 97.88 | 90.34 | 67.99 |
| (RL) Densenet121-pt | 99.90 | 95.94 | 78.70 |
| (RL) Densenet121-adv | 97.07 | 90.59 | 68.16 |
| (RL) Densenet121-pt-adv | 99.38 | 95.26 | 73.56 |

Table 6: Transfer analysis in CIFAR-10 datasets on 4-layer-CNN, evaluation under PGD-250, $\ell_2$, $\epsilon = 7$, step size $\alpha = 0.3$, non-targeted.

| Model | (SL) 4-layer-CNN | (SL) 4-layer-CNN-adv | (RL) 4-layer-CNN | (RL) 4-layer-CNN-adv |
|---|---|---|---|---|
| (SL) 4-layer-CNN | 5.89 | 8.20 | 34.79 | 35.20 |
| (SL) 4-layer-CNN-adv | 6.82 | 6.95 | 35.33 | 35.66 |
| (RL) 4-layer-CNN | 7.25 | 8.81 | 35.21 | 34.48 |
| (RL) 4-layer-CNN-adv | 13.09 | 14.40 | 34.99 | 33.74 |
| (SL) 6-layer-CNN | 14.09 | 14.27 | 35.20 | 35.30 |
| (SL) 6-layer-CNN-adv | 15.16 | 15.87 | 36.48 | 37.07 |
| (RL) 6-layer-CNN | 13.44 | 13.58 | 35.16 | 35.57 |
| (RL) 6-layer-CNN-adv | 32.24 | 33.48 | 47.53 | 45.59 |

Table 7: Transfer analysis in CIFAR-10 datasets on 6-layer-CNN, evaluation under PGD-250, $\ell_2$, $\epsilon = 7$, step size $\alpha = 0.3$, non-targeted.

| Model | (SL) 6-layer-CNN | (SL) 6-layer-CNN-adv | (RL) 6-layer-CNN | (RL) 6-layer-CNN-adv |
|---|---|---|---|---|
| (SL) 6-layer-CNN | 5.81 | 7.18 | 48.45 | 43.15 |
| (SL) 6-layer-CNN-adv | 9.53 | 5.74 | 50.14 | 44.86 |
| (RL) 6-layer-CNN | 8.29 | 7.92 | 48.84 | 42.69 |
| (RL) 6-layer-CNN-adv | 30.88 | 22.56 | 54.31 | 46.91 |
| (SL) 4-layer-CNN | 37.40 | 30.03 | 56.19 | 50.09 |
| (SL) 4-layer-CNN-adv | 36.32 | 29.19 | 56.00 | 50.51 |
| (RL) 4-layer-CNN | 32.23 | 25.94 | 56.65 | 49.92 |
| (RL) 4-layer-CNN-adv | 38.86 | 31.22 | 56.54 | 49.30 |

Table 8: Transfer analysis in CIFAR-10 datasets on Resnet18, evaluation under PGD-250, $\ell_2$, $\epsilon = 7$, step size $\alpha = 0.3$, non-targeted.

| Model | (SL) Resnet18 | (SL) Resnet18-adv | (RL) Resnet18 | (RL) Resnet18-adv | (SL) Resnet18-pt | (SL) Resnet18-pt-adv | (RL) Resnet18-pt | (RL) Resnet18-pt-adv |
|---|---|---|---|---|---|---|---|---|
| (SL) Resnet18 | 46.08 | 65.09 | 81.07 | 89.54 | 88.47 | 86.94 | 88.95 | 89.70 |
| (SL) Resnet18-adv | 80.05 | 28.47 | 85.14 | 89.01 | 88.71 | 87.34 | 89.03 | 89.47 |
| (RL) Resnet18 | 66.93 | 63.77 | 75.44 | 86.82 | 84.99 | 82.69 | 85.59 | 87.52 |
| (RL) Resnet18-adv | 81.89 | 75.34 | 80.04 | 70.09 | 87.05 | 83.30 | 87.40 | 81.40 |
| (SL) Resnet18-pt | 84.65 | 84.36 | 83.04 | 93.80 | 67.61 | 58.02 | 65.53 | 84.48 |
| (SL) Resnet18-pt-adv | 90.26 | 85.97 | 89.19 | 93.68 | 73.84 | 56.96 | 77.56 | 88.86 |
| (RL) Resnet18-pt | 82.40 | 81.23 | 82.40 | 91.68 | 67.68 | 59.25 | 65.07 | 79.69 |
| (RL) Resnet18-pt-adv | 85.05 | 83.65 | 87.37 | 71.23 | 75.32 | 65.08 | 73.26 | 68.40 |
| (SL) Densenet121 | 74.63 | 70.23 | 81.14 | 86.78 | 85.78 | 84.52 | 86.37 | 86.73 |
| (SL) Densenet121-adv | 81.55 | 74.77 | 85.23 | 88.04 | 87.73 | 86.58 | 87.90 | 88.30 |
| (RL) Densenet121 | 69.87 | 66.20 | 77.10 | 86.18 | 84.48 | 82.42 | 85.02 | 86.51 |
| (RL) Densenet121-adv | 80.64 | 76.51 | 82.31 | 80.99 | 84.83 | 82.05 | 84.88 | 83.38 |
| (SL) Densenet121-pt | 90.58 | 89.67 | 87.76 | 95.32 | 79.11 | 78.97 | 82.88 | 93.26 |
| (SL) Densenet121-pt-adv | 92.43 | 88.91 | 90.38 | 94.42 | 86.72 | 74.00 | 89.02 | 92.62 |
| (RL) Densenet121-pt | 86.13 | 85.57 | 83.82 | 94.11 | 79.46 | 78.57 | 83.76 | 92.56 |
| (RL) Densenet121-pt-adv | 90.37 | 88.46 | 90.00 | 77.53 | 86.52 | 78.74 | 87.71 | 70.56 |

Table 9: Transfer analysis in CIFAR-10 datasets on Densenet121, evaluation under PGD-250, $\ell_2$, $\epsilon = 7$, step size $\alpha = 0.3$, non-targeted.

| Model | (SL) Densenet121 | (SL) Densenet121-adv | (RL) Densenet121 | (RL) Densenet121-adv | (SL) Densenet121-pt | (SL) Densenet121-pt-adv | (RL) Densenet121-pt | (RL) Densenet121-pt-adv |
|---|---|---|---|---|---|---|---|---|
| (SL) Densenet121 | 31.33 | 84.01 | 84.84 | 86.24 | 85.72 | 84.34 | 86.05 | 86.60 |
| (SL) Densenet121-adv | 87.42 | 15.77 | 87.19 | 88.00 | 87.72 | 86.33 | 88.25 | 88.26 |
| (RL) Densenet121 | 83.93 | 82.56 | 67.99 | 79.20 | 84.26 | 81.68 | 84.83 | 86.01 |
| (RL) Densenet121-adv | 84.19 | 83.85 | 79.19 | 68.16 | 84.41 | 82.35 | 84.81 | 83.83 |
| (SL) Densenet121-pt | 95.59 | 95.51 | 94.68 | 95.29 | 52.00 | 41.56 | 79.32 | 91.91 |
| (SL) Densenet121-pt-adv | 95.70 | 95.41 | 95.11 | 95.38 | 72.97 | 28.80 | 82.56 | 90.13 |
| (RL) Densenet121-pt | 93.66 | 93.54 | 92.43 | 93.55 | 65.12 | 53.38 | 78.70 | 90.50 |
| (RL) Densenet121-pt-adv | 93.05 | 92.89 | 92.48 | 89.76 | 86.28 | 69.96 | 86.87 | 73.56 |
| (SL) Resnet18 | 88.50 | 87.78 | 87.97 | 89.12 | 88.64 | 87.11 | 88.91 | 89.57 |
| (SL) Resnet18-adv | 88.74 | 87.97 | 88.27 | 88.97 | 88.64 | 87.73 | 88.88 | 89.33 |
| (RL) Resnet18 | 85.73 | 85.25 | 83.88 | 86.43 | 85.07 | 82.75 | 85.54 | 87.43 |
| (RL) Resnet18-adv | 87.19 | 86.91 | 87.10 | 84.96 | 86.79 | 83.78 | 87.57 | 83.26 |
| (SL) Resnet18-pt | 94.40 | 94.14 | 92.53 | 93.97 | 82.53 | 74.45 | 84.85 | 92.86 |
| (SL) Resnet18-pt-adv | 94.62 | 94.22 | 94.04 | 94.42 | 90.51 | 80.01 | 90.67 | 92.90 |
| (RL) Resnet18-pt | 91.69 | 91.58 | 90.04 | 91.62 | 81.57 | 73.85 | 84.08 | 90.24 |
| (RL) Resnet18-pt-adv | 89.15 | 89.41 | 88.78 | 83.15 | 85.61 | 73.07 | 87.38 | 73.35 |

Table 10: Model robustness evaluation (train, test, and AEs) in CIFAR-100 datasets across models, evaluation under PGD-250, $\ell_2$, $\epsilon = 7$, step size $\alpha = 0.3$, non-targeted.

| Model | Clean test (%) | AE (%) |
|---|---|---|
| (SL) 4-layer-CNN | 52.40 | 3.80 |
| (SL) 4-layer-CNN-adv | 50.77 | 4.06 |
| (RL) 4-layer-CNN | 28.38 | 15.06 |
| (RL) 4-layer-CNN-adv | 29.89 | 17.13 |
| (SL) 6-layer-CNN | 64.75 | 2.53 |
| (SL) 6-layer-CNN-adv | 63.61 | 2.83 |
| (RL) 6-layer-CNN | 59.80 | 13.06 |
| (RL) 6-layer-CNN-adv | 56.54 | 25.51 |
| (SL) Resnet18 | 69.36 | 14.83 |
| (SL) Resnet18-pt | 80.76 | 30.65 |
| (SL) Resnet18-adv | 68.76 | 12.61 |
| (SL) Resnet18-pt-adv | 79.70 | 26.55 |
| (RL) Resnet18 | 67.08 | 32.15 |
| (RL) Resnet18-pt | 77.22 | 45.91 |
| (RL) Resnet18-adv | 65.07 | 29.51 |
| (RL) Resnet18-pt-adv | 74.68 | 41.93 |
| (SL) Densenet121 | 66.12 | 13.11 |
| (SL) Densenet121-pt | 83.49 | 13.92 |
| (SL) Densenet121-adv | 65.86 | 12.20 |
| (SL) Densenet121-pt-adv | 82.64 | 11.46 |
| (RL) Densenet121 | 65.39 | 33.68 |
| (RL) Densenet121-pt | 80.54 | 34.97 |
| (RL) Densenet121-adv | 64.05 | 34.20 |
| (RL) Densenet121-pt-adv | 77.64 | 31.02 |

Table 11: Transfer analysis in CIFAR-100 datasets on 4-layer-CNN, evaluation under PGD-250, $\ell_2$, $\epsilon = 7$, step size $\alpha = 0.3$, non-targeted.

| Model | (SL) 4-layer-CNN | (SL) 4-layer-CNN-adv | (RL) 4-layer-CNN | (RL) 4-layer-CNN-adv |
|---|---|---|---|---|
| (SL) 4-layer-CNN | 0.02 | 1.91 | 21.65 | 23.74 |
| (SL) 4-layer-CNN-adv | 3.16 | 0.04 | 23.54 | 25.18 |
| (RL) 4-layer-CNN | 5.37 | 6.17 | 10.48 | 11.89 |
| (RL) 4-layer-CNN-adv | 5.68 | 6.26 | 10.81 | 12.40 |
| (SL) 6-layer-CNN | 7.47 | 9.47 | 27.57 | 26.62 |
| (SL) 6-layer-CNN-adv | 7.85 | 9.93 | 27.82 | 27.03 |
| (RL) 6-layer-CNN | 5.69 | 8.30 | 24.16 | 23.64 |
| (RL) 6-layer-CNN-adv | 15.60 | 16.73 | 30.88 | 30.10 |

Table 12: Transfer analysis in CIFAR-100 datasets on 6-layer-CNN, evaluation under PGD-250, $\ell_2$, $\epsilon = 7$, step size $\alpha = 0.3$, non-targeted.

| Model | (SL) 6-layer-CNN | (SL) 6-layer-CNN-adv | (RL) 6-layer-CNN | (RL) 6-layer-CNN-adv |
|---|---|---|---|---|
| (SL) 6-layer-CNN | 0.03 | 0.40 | 8.27 | 16.59 |
| (SL) 6-layer-CNN-adv | 1.32 | 0.07 | 9.09 | 17.78 |
| (RL) 6-layer-CNN | 1.82 | 1.30 | 8.04 | 15.89 |
| (RL) 6-layer-CNN-adv | 20.17 | 14.52 | 13.08 | 18.16 |
| (SL) 4-layer-CNN | 30.50 | 27.22 | 30.70 | 29.60 |
| (SL) 4-layer-CNN-adv | 33.34 | 30.23 | 33.76 | 30.53 |
| (RL) 4-layer-CNN | 21.63 | 20.61 | 21.83 | 21.16 |
| (RL) 4-layer-CNN-adv | 21.50 | 19.63 | 21.87 | 20.35 |

Table 13: Transfer analysis in CIFAR-100 datasets on Resnet18, evaluation under PGD-250, $\ell_2$, $\epsilon = 7$, step size $\alpha = 0.3$, non-targeted.

| Model | (SL) Resnet18 | (SL) Resnet18-adv | (RL) Resnet18 | (RL) Resnet18-adv | (SL) Resnet18-pt | (SL) Resnet18-pt-adv | (RL) Resnet18-pt | (RL) Resnet18-pt-adv |
|---|---|---|---|---|---|---|---|---|
| (SL) Resnet18 | 14.83 | 59.61 | 51.09 | 56.82 | 69.70 | 68.99 | 66.98 | 67.98 |
| (SL) Resnet18-adv | 52.08 | 12.61 | 48.84 | 54.36 | 68.67 | 67.42 | 65.63 | 65.37 |
| (RL) Resnet18 | 51.47 | 57.22 | 32.15 | 48.92 | 63.71 | 63.12 | 59.86 | 67.31 |
| (RL) Resnet18-adv | 65.37 | 65.99 | 60.73 | 29.51 | 76.06 | 75.00 | 73.27 | 43.57 |
| (SL) Resnet18-pt | 65.23 | 65.76 | 62.06 | 59.95 | 30.65 | 31.36 | 41.02 | 56.81 |
| (SL) Resnet18-pt-adv | 64.97 | 65.74 | 62.02 | 60.25 | 27.70 | 26.55 | 41.23 | 56.95 |
| (RL) Resnet18-pt | 65.13 | 65.89 | 61.98 | 60.27 | 47.20 | 47.26 | 45.91 | 58.98 |
| (RL) Resnet18-pt-adv | 66.23 | 66.54 | 63.46 | 59.32 | 71.38 | 69.74 | 68.36 | 41.93 |
| (SL) Densenet121 | 65.73 | 66.37 | 62.86 | 60.75 | 77.94 | 76.55 | 74.08 | 70.57 |
| (SL) Densenet121-adv | 65.04 | 65.98 | 62.65 | 60.09 | 77.19 | 76.79 | 73.99 | 69.71 |
| (RL) Densenet121 | 63.86 | 65.28 | 60.46 | 59.85 | 76.24 | 75.34 | 73.15 | 69.33 |
| (RL) Densenet121-adv | 65.43 | 66.02 | 62.11 | 59.30 | 77.58 | 76.44 | 73.92 | 65.51 |
| (SL) Densenet121-pt | 64.87 | 65.62 | 61.71 | 59.98 | 62.41 | 61.34 | 64.30 | 63.80 |
| (SL) Densenet121-pt-adv | 64.97 | 65.86 | 61.95 | 60.01 | 63.12 | 61.58 | 64.62 | 63.59 |
| (RL) Densenet121-pt | 64.92 | 65.63 | 61.49 | 60.18 | 62.40 | 61.57 | 62.85 | 66.05 |
| (RL) Densenet121-pt-adv | 65.96 | 66.43 | 63.35 | 59.24 | 75.20 | 74.34 | 72.83 | 39.58 |

Table 14: Transfer analysis in CIFAR-100 datasets on Densenet121, evaluation under PGD-250, $\ell_2$, $\epsilon = 7$, step size $\alpha = 0.3$, non-targeted.

| Model | (SL) Densenet121 | (SL) Densenet121-adv | (RL) Densenet121 | (RL) Densenet121-adv | (SL) Densenet121-pt | (SL) Densenet121-pt-adv | (RL) Densenet121-pt | (RL) Densenet121-pt-adv |
|---|---|---|---|---|---|---|---|---|
| (SL) Densenet121 | 13.11 | 62.69 | 60.26 | 56.20 | 80.67 | 79.11 | 76.71 | 70.73 |
| (SL) Densenet121-adv | 57.72 | 12.20 | 59.74 | 55.60 | 80.64 | 78.98 | 76.46 | 70.35 |
| (RL) Densenet121 | 56.74 | 61.67 | 33.68 | 50.26 | 79.00 | 77.39 | 74.98 | 70.11 |
| (RL) Densenet121-adv | 58.94 | 62.65 | 52.22 | 34.20 | 80.52 | 78.79 | 76.66 | 64.78 |
| (SL) Densenet121-pt | 57.95 | 62.34 | 59.66 | 55.91 | 13.92 | 25.47 | 41.71 | 58.06 |
| (SL) Densenet121-pt-adv | 57.73 | 62.29 | 59.61 | 55.84 | 23.87 | 11.46 | 43.05 | 59.39 |
| (RL) Densenet121-pt | 57.89 | 62.55 | 59.79 | 56.02 | 42.08 | 42.95 | 34.97 | 60.77 |
| (RL) Densenet121-pt-adv | 59.24 | 63.09 | 60.90 | 55.58 | 75.96 | 74.31 | 72.38 | 31.02 |
| (SL) Resnet18 | 49.94 | 57.09 | 53.72 | 53.45 | 74.22 | 72.90 | 69.93 | 67.73 |
| (SL) Resnet18-adv | 48.58 | 52.94 | 51.64 | 51.24 | 73.44 | 72.19 | 69.57 | 66.68 |
| (RL) Resnet18 | 48.79 | 55.41 | 49.89 | 53.12 | 69.30 | 68.06 | 64.66 | 68.47 |
| (RL) Resnet18-adv | 58.10 | 62.34 | 60.14 | 52.03 | 79.48 | 77.82 | 76.09 | 46.76 |
| (SL) Resnet18-pt | 58.13 | 62.38 | 59.98 | 56.17 | 57.63 | 57.91 | 60.71 | 58.87 |
| (SL) Resnet18-pt-adv | 58.26 | 62.56 | 60.03 | 56.06 | 59.10 | 59.06 | 62.09 | 58.57 |
| (RL) Resnet18-pt | 58.42 | 62.85 | 59.98 | 56.33 | 61.57 | 61.49 | 61.63 | 62.83 |
| (RL) Resnet18-pt-adv | 59.40 | 63.13 | 60.91 | 56.00 | 78.67 | 77.38 | 75.69 | 44.56 |

Table 15: Model robustness evaluation (train, test, and AEs) in ImageNet-100 datasets across models, evaluation under PGD-250, $\ell_2$, $\epsilon = 7$, step size $\alpha = 0.3$, non-targeted.

| Model | Clean test (%) | AE (%) |
|---|---|---|
| (SL) 4-layer-CNN | 48.58 | 2.80 |
| (SL) 4-layer-CNN-adv | 45.76 | 2.92 |
| (RL) 4-layer-CNN | 47.88 | 10.20 |
| (RL) 4-layer-CNN-adv | 47.56 | 11.06 |
| (SL) 6-layer-CNN | 57.64 | 5.72 |
| (SL) 6-layer-CNN-adv | 58.00 | 5.36 |
| (RL) 6-layer-CNN | 55.60 | 18.04 |
| (RL) 6-layer-CNN-adv | 45.92 | 18.24 |
| (SL) Resnet18 | 74.00 | 45.00 |
| (SL) Resnet18-adv | 74.90 | 42.62 |
| (RL) Resnet18 | 73.92 | 49.02 |
| (RL) Resnet18-adv | 65.28 | 42.96 |

### A.8.4 PLACES-365 (EXTRA)

The performance on Places-365 across 4-layer CNN, 6-layer CNN, and ResNet-18 is shown in Table 16, which is not discussed in the main paper due to incomplete experiments. This incomplete experiment is because reinforcement-learning-based–based training is computationally prohibitive for Places-365 under our experimental settings. For example, training one RL model in our settings on a single NVIDIA A100 is estimated to take multiple months. This indicates that improving the efficiency and scalability of the RL training pipeline will be an important direction for future work.

Table 16: Model robustness evaluation (train, test, and AEs) in Places-365 datasets across models, evaluation under PGD-250, $\ell_2$, $\epsilon = 7$, step size $\alpha = 0.3$, non-targeted.

| Model | Clean test (%) | AE (%) |
|---|---|---|
| (SL) 4-layer-CNN | 28.86 | 13.01 |
| (SL) 4-layer-CNN-adv | 29.79 | 9.96 |
| (RL) 4-layer-CNN | - | - |
| (RL) 4-layer-CNN-adv | - | - |
| (SL) 6-layer-CNN | 34.72 | 5.86 |
| (SL) 6-layer-CNN-adv | 35.53 | 6.58 |
| (RL) 6-layer-CNN | - | - |
| (RL) 6-layer-CNN-adv | - | - |
| (SL) Resnet18 | 42.93 | - |
| (SL) Resnet18-adv | 43.99 | - |
| (RL) Resnet18 | - | - |
| (RL) Resnet18-adv | - | - |

### A.9 THEORETICAL ANALYSIS FOR REINFORCEMENT LEARNING ROBUSTNESS

**Assumption 1 (Empirical gradient property).** As observed in Section 6.2 and Section 6.3, RL training tends to induce flatter loss landscapes than SL, i.e., $\|\nabla_x \hat{f}_{\mathrm{RL}}(x)\|_2 \leq \|\nabla_x \hat{f}_{\mathrm{SL}}(x)\|_2$ holds approximately on average over the data distribution.

**Proposition 1 (First-order influence bound).** Under Assumption 1, for a neural network $\hat{f} : \mathcal{X} \to \mathbb{R}^C$ and an adversarial perturbation $\beta \in \mathcal{B}_\epsilon := \{\beta \in \mathbb{R}^d : \|\beta\|_2 \leq \epsilon\}$, the expected first-order output change satisfies

$$\mathbb{E}_{x\sim\mathcal{D}}[\Delta \hat{f}_{\mathrm{RL}}(x)] \leq \mathbb{E}_{x\sim\mathcal{D}}[\Delta \hat{f}_{\mathrm{SL}}(x)], \quad \Delta\hat{f}(x) := \hat{f}(x+\beta) - \hat{f}(x).$$

**Proof sketch.**

1. *First-Order Taylor Expansion* (small $\epsilon$):

$\Delta \hat{f}(x) = \nabla_x \hat{f}(x)^\top \beta + \mathcal{O}(\epsilon^2)$.

2. *Gradient Norm Bound*:

$\mathbb{E}_{x \sim \mathcal{D}}\big[\|\nabla_x \hat{f}_{\mathrm{RL}}(x)\|_2\big] \leq \mathbb{E}_{x \sim \mathcal{D}}\big[\|\nabla_x \hat{f}_{\mathrm{SL}}(x)\|_2\big]$.

3. *Expectation Transformation*:

$$\mathbb{E}_{x \sim \mathcal{D}}\left[\Delta \hat{f}(x)\right] = \mathbb{E}_{x \sim \mathcal{D}}\left[\hat{f}(x+\beta) - \hat{f}(x)\right]$$

$$\approx \mathbb{E}_{x \sim \mathcal{D}}\left[\nabla_x \hat{f}(x)^\top \beta\right] \quad \text{(First-order Taylor approximation)}$$

$$\leq \mathbb{E}_{x \sim \mathcal{D}}\left[\|\nabla_x \hat{f}(x)\|_2 \cdot \|\beta\|_2\right] \quad \text{(Cauchy-Schwarz inequality)}$$

$$= \mathbb{E}_{x \sim \mathcal{D}}\left[\|\nabla_x \hat{f}(x)\|_2\right] \cdot \|\beta\|_2 \quad (\|\beta\|_2 \text{ is constant)}.$$

After applying $\|\nabla_x \hat{f}_{\mathrm{RL}}(x)\|_2 \leq \|\nabla_x \hat{f}_{\mathrm{SL}}(x)\|_2$:

We obtain:

$$\mathbb{E}[\Delta \hat{f}_{\mathrm{RL}}(x)] \leq \underbrace{\mathbb{E}[\|\nabla_x \hat{f}_{\mathrm{RL}}(x)\|_2]}_{\text{Smaller}} \cdot \|\beta\|_2 \leq \underbrace{\mathbb{E}[\|\nabla_x \hat{f}_{\mathrm{SL}}(x)\|_2]}_{\text{Larger}} \cdot \|\beta\|_2 \leq \mathbb{E}[\Delta \hat{f}_{\mathrm{SL}}(x)] \,.$$

**Corollary 1.1**. *In the small-$\epsilon$ regime, RL models exhibit greater adversarial robustness as their predictions are less sensitive to perturbations compared to SL models.*

*Remark.* This proposition should be interpreted as a first-order sensitivity analysis *conditioned on* the empirically observed gradient property in Assumption 1, rather than as an unconditional robustness guarantee for arbitrary architectures.

## A.10 LLM USAGE DISCLOSURE

We used Large Language Models (LLMs) (shown in Table 17) **only for English-language polishing**, including grammar correction and wording suggestions. **No new scientific contents, including equations, experimental designs, or codes** were generated by the models. We did not provide any non-public data or code to the models. All model suggestions were manually reviewed and edited by the authors for technical correctness. The authors take full responsibility for the final content; the LLM is not an author.

Table 17: LLM usage disclosure.

| Model | Version | Access date |
|---|---|---|
| ChatGPT | GPT-5 (Auto) | 2025.07–2025.09 |
| ChatGPT | GPT-5 (Thinking) | 2025.07–2025.09 |
| Claude | Sonnet 4 | 2025.07–2025.09 |
| ChatGPT | GPT4o | 2024.12–2025.03 |

## A.11 ADVERSARIAL ATTACK: AUTOATTACK EVALUATION ON CNN BACKBONES

In addition to evaluating robustness under standard PGD attacks, we further adopt AutoAttack as a complementary and more reliable assessment of adversarial robustness. For experiments on CIFAR-10, CIFAR-100, and ImageNet-100, we set the maximum perturbation budgets for AutoAttack to $\epsilon = 7, 7$, and $3.5$, respectively, using the standard AutoAttack configuration. The full AutoAttack evaluation results for the 6-layer CNN on CIFAR-100 and ImageNet-100 are provided in Table 18 and Table 19. In both cases, CNN-RL-adv achieves the strongest robustness, followed by CNN-SL-adv, while the non-adversarially trained variants (CNN-RL and CNN-SL) show similarly low resistance to AutoAttack.

Table 18: AutoAttack results for CNN on CIFAR-100 dataset.

|  | Clean (%) | APGD-CE (%) | APGD-T (%) | FAB-T (%) | SQUARE (%) |
|---|---|---|---|---|---|
| 6-layer-CNN-SL | 64.75 | 4.75 | 3.91 | 3.91 | 3.48 |
| 6-layer-CNN-SL-adv | 63.3 | 19.02 | 15.91 | 15.91 | 13.95 |
| 6-layer-CNN-RL | 59.8 | 4.19 | 3.44 | 3.44 | 3.04 |
| 6-layer-CNN-RL-adv | 56.54 | 17.95 | 17.31 | 17.31 | 15.14 |

Table 19: AutoAttack results for CNN on CIFAR-100 dataset.

|  | Clean (%) | APGD-CE (%) | APGD-T (%) | FAB-T (%) | SQUARE (%) |
|---|---|---|---|---|---|
| 6-layer-CNN-SL | 57.64 | 5.64 | 4.24 | 4.24 | 3.62 |
| 6-layer-CNN-SL-adv | 42.08 | 11.54 | 9.34 | 9.34 | 8.08 |
| 6-layer-CNN-RL | 55.6 | 4.96 | 3.52 | 3.52 | 3.1 |
| 6-layer-CNN-RL-adv | 45.92 | 13.08 | 10.72 | 10.72 | 8.94 |

## A.12 PARETO ANALYSIS: CLEAN-ROBUST TRADE-OFF

To investigate the trade-off between clean accuracy and adversarial robustness, we conduct a trade-off evaluation using a 6-layer CNN architecture on CIFAR-10. We compare Supervised Learning (SL) and Reinforcement Learning (RL) under identical training settings, including a learning rate of $1 \times 10^{-4}$ and a batch size of 256. To account for optimization stochasticity, we train both RL-adv and SL with three independent random seeds $\{10, 20, 42\}$, while the seed for the standard PGD attack is fixed to 33. We report the mean clean accuracy and robust accuracy (mean $\pm$ standard deviation across seeds). Robustness is evaluated using a standard PGD, ensuring a consistent evaluation protocol.

