# OpenReview forum: "Can Exploration Save Us from Adversarial Attacks? A Reinforcement Learning Approach to Adversarial Robustness"
_ICLR.cc/2026/Conference — Submitted to ICLR 2026_

### Official Review · Reviewer_oqud · 2025-10-29

**Soundness:** 4
**Presentation:** 4
**Contribution:** 3
**Rating:** 8
**Confidence:** 3

**Summary:**

- Casts image classification as a one-step RL problem and trains classifiers as policies with ε-greedy exploration (optionally combined with FGSM/TRADES-style regularization).

- Reports gains in white-box L2 PGD robustness on CIFAR-10/100 and ImageNet-100, with a modest drop in clean accuracy versus standard supervised training.

- Observes transfer asymmetry: adversarial examples crafted on RL models transfer poorly, while those crafted on SL models still hurt plain RL

- RL training yields smaller gradient norms and less stable input-gradient directions, making gradient-based attacks less effective.

- Acknowledges limitations: evaluation lacks stronger standardized attacks (e.g., AutoAttack) and fuller threat models; training introduces extra compute overhead.

**Strengths:**

- Clear empirical finding: ε-greedy RL training notably boosts white-box L2 PGD robustness with modest clean-accuracy drop

- Solid diagnostics: analyzes gradient norms, direction stability, and loss landscape to explain why attacks struggle

- Broad(ish) coverage: multiple datasets (CIFAR-10/100, ImageNet-100) and several backbones show the effect is not a one-off

- Practical angle: simple to add on top of standard training and compatible with adversarial regularization (e.g., TRADES)

- Transparent about limits: explicitly discusses stronger attacks needed and positions work as complementary, not a silver bullet

**Weaknesses:**

- Evaluation is narrow: no AutoAttack, L∞, or true black-box attacks to rule out gradient masking.

- Transfer remains a problem: adversarial examples crafted on SL models still break plain RL; robustness largely relies on RL + adversarial training.

- Compute/efficiency unclear: added exploration and optional adversarial phase increase cost without a compute-matched analysis

- Scalability is uncertain: results are mostly on smaller backbones/datasets; unclear behavior on modern large ViTs/foundation models.

**Questions:**

- The authors have mentioned the computational overhead, but can you share some stats about this and how it differs from the CNN backbones to ViTs

- I'm really curious to see how this would scale to bigger models

- Also some analysis on the representation space of the model trained with RL and adversarial defense, e.g, how the features could transform to other domain or even their visualization would be nice

---

> ### Author Response · Authors · 2025-11-23
> **Respond to Reviewer oqud (Part 1 of 2)**
>
> Thank you for your excellent and constructive review. We sincerely appreciate your supportive assessment of our work. Below, we provide detailed responses to your questions.
>
> **Computational Overhead.** RL involves exploration in policy optimization, which increases the required number of steps (or epochs) to achieve convergence compared to SL. When using the same per-epoch compute budget and implementation settings, SL typically converges in fewer epochs. This characteristic is consistent across CNN backbones and indicates that scaling RL-based classifiers mainly requires additional compute investment. For reference: under the same hardware (A100 GPU) and for comparable clean accuracy on CIFAR-10, the training duration for a 6-layer CNN increases from approximately 0.5–1 hour (SL) to 2–3 hours (RL). Similar overhead ratios are observed on CIFAR-100 and across CNN backbones. We appreciate the reviewer’s interest in deployment practicality and view improving training efficiency as an orthogonal direction that can further strengthen our approach.
>
> **Applicability to ViT Architectures.** We agree that Vision Transformers (ViTs) are an important direction. We did not include them in the current transfer matrices for specific methodological reasons:
> - **Architectural Consistency:** Our study currently focuses on the *convolutional inductive bias*. ResNet and DenseNet share this bias, allowing for controlled transferability comparisons.
> - **Input Representation:** ViTs fundamentally change input processing (patch tokenization vs. pixel-wise convolution). Directly transferring adversarial examples between these paradigms introduces confounding factors (e.g., resizing artifacts for fixed-size patches) that complicate the isolation of the "RL effect."
>
> However, our RL framework is architecture-agnostic. Applying it to ViTs (e.g., DeiT) is a feasible and exciting avenue for future work to see if the gradient smoothing effect persists in attention-based mechanisms.
>
>
> **ResNet18, ResNet50, DenseNet121.** Thank you for your interest in scalability. During the rebuttal period, we have re-analyzed the results for ResNet-18 and DenseNet-121 and added new experiments on ResNet-50. We observe that the robustness gain from RL persists across widely used CNN architectures:
>
> - CIFAR-10
>     - ResNet-18: RL/adv $\approx$ 70% $>$ SL/adv $\approx$ 30%
>     - ResNet-50: RL/adv $\approx$ 70% $>$ SL/adv $\approx$ 40%
>     - DenseNet-121: RL/adv $\approx$ 65% $>$ SL/adv $\approx$ 30%
> - CIFAR-100
>     - ResNet-18: RL/adv $\approx$ 30% $>$ SL/adv $\approx$ 13%
>     - DenseNet-121: RL/adv $\approx$ 35% $>$ SL/adv $\approx$ 13%
>
> These findings demonstrate that the observed robustness improvement is consistent across different convolutional backbones. Full updated results are included in the Appendix.
>
> **Representation Space of Feature Layers.** We appreciate this insightful suggestion. In the current paper, we have already examined the robustness of representations at the input level (IGV, dIGV, AGN) and at the output level (decision boundary smoothness and predictive uncertainty). We fully agree that analyzing intermediate feature representations could provide additional insight into how RL optimization shapes the robustness of hierarchical abstractions. Such analysis would naturally complement our existing input–feature–output robustness characterization. We are excited about its potential to further enhance interpretability in future work.

---

> > ### Author Response · Authors · 2025-11-23
> > **Respond to Reviewer oqud (Part 2 of 2)**
> >
> > **AutoAttack.** To further strengthen our evaluation, we additionally performed AutoAttack on CIFAR-10, CIFAR-100, and ImageNet-100 using CNN models during the rebuttal period. CNN-RL-adv consistently achieves the highest adversarial accuracy across all four AutoAttack components (APGD-CE, APGD-T, FAB-T, and SQUARE), while maintaining comparable clean accuracy. Similar improvements are observed on CIFAR-100 and ImageNet-100, confirming that RL-based optimization strengthens robustness even under strong standardized evaluation.
> >
> > **Table 1: AutoAttack results for CNN on CIFAR-10 dataset:**
> > | Model       | Clean (%)  | APGD-CE (%) | APGD-T (%) | FAB-T (%)  | SQUARE (%) |
> > | :---         | :----:     | :----:      | :----: | :----: | :----: |
> > | CNN-SFT      | **90.82\***  | 15.42       | 13.58  | 13.58  | 11.6   |
> > | CNN-SFT-adv  | 87.03      | 24.87       | 21.77  | 21.77  | 19.55  |
> > | CNN-RL       | 88.62      | 16.71       | 13.73  | 13.73  | 11.96  |
> > | CNN-RL-adv   | 86.29      | **36.27\***   | **35.4\***   | **35.4\***   | **32.41\***  |
> >
> > **Table 2: AutoAttack results for CNN on CIFAR-100 dataset:**
> > | Model       | Clean (%)  | APGD-CE (%) | APGD-T (%) | FAB-T (%)  | SQUARE (%) |
> > | :---         | :----:     | :----:      | :----: | :----: | :----: |
> > | CNN-SFT      | **64.75\***  | 4.75       | 3.91  | 3.91  | 3.48  |
> > | CNN-SFT-adv  | 63.3      | **19.02\***       | 15.91  | 15.91  | 13.95  |
> > | CNN-RL       | 59.8      | 4.19       | 3.44  | 3.44  | 3.04  |
> > | CNN-RL-adv   | 56.54     | 17.95   | **17.31\***   | **17.31\***   | **15.14\***  |
> >
> > **Table 3: AutoAttack results for CNN on ImageNet-100 dataset:**
> > | Model       | Clean (%)  | APGD-CE (%) | APGD-T (%) | FAB-T (%)  | SQUARE (%) |
> > | :---         | :----:     | :----:      | :----: | :----: | :----: |
> > | CNN-SFT      | **57.64\***  | 5.64       | 4.24  | 4.24  | 3.62  |
> > | CNN-SFT-adv  | 42.08      | 11.54       | 9.34  | 9.34  | 8.08 |
> > | CNN-RL       | 55.6      | 4.96      | 3.52  | 3.52  | 3.1  |
> > | CNN-RL-adv   | 45.92     | **13.08\***   | **10.72\***   | **10.72\***   | **8.94\***  |

---

> > > ### Comment · Reviewer_oqud · 2025-11-27
> > >
> > > Thank you for your detailed response to my comments; I will maintain my positive recommendation.

---

### Official Review · Reviewer_GToB · 2025-10-31

**Soundness:** 2
**Presentation:** 1
**Contribution:** 1
**Rating:** 2
**Confidence:** 5

**Summary:**

This paper provides a rarely explored perspective of combing reinforcement learning (RL) with adversarial training (AT) for more robust image classification. Experiments are conducted on 4/6-layer CNN and ResNet-18.

**Strengths:**

Figure.3 is interesting and intuitive.

**Weaknesses:**

Overall, this paper is largely unpolished and barely meets the standard for a readable scientific paper. Major issues range from motivation, novelty, solidity and clarity.

**Motivation**. Since AT is notoriously well-known for its degradation on clean performance and prolonged training time, combining it with RL, which is also extremely expensive for training, is counter-intuitive and requires a strong motivation. However, this paper seems to be merely motivated by relative less exploration, making one question why taking the labour of RL to combine it with AT for even worse efficiency and performance only for boosted robustness.

**Novelty**. Furthermore, according to sec.4, the entire method is build upon simple combination of existing $\epsilon$-greedy RL and TRADES. The paper itself provides little or no insights/innovation beyond a simple 1+1. Besides, the preliminary feels remotely related to the upcoming method section,  making no contribution to understanding the proposed method.

**Solidity**. Experiments are conducted on severely subpar models, i.e., CNN and RN18, without an explicit explanation but an adjective ‘representative’. No mainstream models, including ViT, Wide ResNet, or even RN-50, are tested. Furthermore, the baselines used for training and attacks are also significantly outdated. No recently SOTA methods such as AutoAttack, are considered or discussed, underming the experimental solidity of the paper.


**Clarity**: Writing-wise, this paper is poorly written and full of format/punctuation/spelling mistakes, further lowering the readability of this paper. To start with, sec.3 seems diverge largely from the storyline, making it confused to those many introduced terms, to which are not referred until the experiment. Besides, repetitive notations also appear during introduction of the method, e.g., $\epsilon$ is used for both RL and AT. For format, most of the reference formats are incorrect, nested and hard to trace. Many references are placed at the end of sentences where no clear clue is given. Caption for tables are incorrectly capitalized, and the top rules of tables are constantly missing. Lastly, for punctuations, all quotation marks are incorrectly used, and all captions of tables are ended without periods.

**Miscellaneous**. I) Although an anonymous GitHub repository is provided claiming to include configuration and codes, the repository contains nothing but the paper abstract. II) It is unclear why the author include several formal definition of existing conceptions/methods into the supplementary.

Overall, this paper falls far behind regarding a standard scientific paper and requires enormous effort to improve its motivation, novelty, solidity and clarity.

**Questions:**

See weaknesses.

---

> ### Author Response · Authors · 2025-11-23
> **Respond to Reviewer GToB**
>
> Thank you for the detailed and candid feedback. We appreciate the opportunity to improve the clarity, organization, and solidity of our manuscript. Below, we respond to your comments point by point.
>
> **Motivation.** We agree that combining AT and RL is computationally expensive. However, our motivation is not to propose a low-cost defense, but to investigate a scientific hypothesis: whether the *exploration mechanism* in RL induces gradient behaviors that may influence adversarial vulnerability. Motivated by robustness phenomena in other RL domains (Xu et al., 2019, LexicalAT), we aim to provide empirical evidence clarifying these mechanisms in image classification. We emphasize that this work is positioned as an exploratory study into robustness mechanisms rather than an efficiency-focused proposal.
>
> **Novelty.** We understand the reviewer’s concern that our method only builds on standard RL with TRADES. However, our contribution is not architectural, but conceptual: we analyze how RL differs from SL in terms of gradient behavior and robustness under adversarial perturbations. Therefore, while the technical components are standard, we believe these analyses offer useful insights for understanding robustness mechanisms.
>
> **Solidity: ResNet-50 and AutoAttack.** We acknowledge the critique regarding model scale and baselines. To address this, we have performed extensive new experiments:
>
> - **Complex Architectures:** We added **ResNet-50** experiments on CIFAR-10. The results indicate that RL-adv consistently outperforms SL-adv, showing that our findings scale to larger models.
> - **Stronger Baselines:** We evaluated our models against **AutoAttack** (Standard and Plus). As detailed in the General Response tables, the RL-based approach maintains its robustness advantage even under these state-of-the-art attacks.
>
> These results have been added to the revised manuscript to strengthen the empirical solidity.
>
> **Clarity.** We agree that this feedback is helpful for improving readability. In the revised version, we address each of the reviewers’ concerns as follows:
>
> 1. **Organization of Section 3**: We have reorganized Section 3 to ensure a smoother flow and a stronger connection with the rest of the paper. At the beginning of the section, we now clearly explain why these preliminaries are needed and how they relate to the subsequent methodological development and experimental analysis.
> 2. **Clarity of the main idea**: We substantially clarified the central message of the paper throughout the abstract and main text. In particular, we now clearly state that the goal of this work is not only to investigate the use of RL for classification under adversarial perturbations, but also to show how our experiments help explain the empirical behaviors we observe under such perturbations. These insights further motivate future directions that combine RL and SL, guided by the analysis presented in the paper, to potentially mitigate adversarial examples while preserving both accuracy and efficiency.
> 3. **Citation**: Some duplicate citations were found due to importing BibTeX entries from different sources (e.g., arXiv and Google Scholar), and we have resolved these issues. Some author names in the references were truncated using “et al.” , and these have been corrected to display the complete author lists. Additionally, we corrected the misuse of "citep" and "citet", and improved the citation formatting (e.g., color and style) to enhance readability.
> 4. **Punctuation**: Missing punctuation in tables and figures has been added. The table formatting (e.g., top rules) and caption capitalization have also been revised to improve clarity and consistency in the context.
> 5. **Appendix**: Several of these issues were in the appendix, which has now been substantially revised.
> 6. **Notation**: The reuse of the symbol $\epsilon$ in different contexts (e.g., $\epsilon$-greedy in RL and perturbation magnitude in adversarial attacks) could cause confusion. Although both usages are standard in their respective areas, we have updated the notation to distinguish them by using '$\varepsilon_{Greedy}$' for $\epsilon$-greedy exploration and '$\epsilon$' for adversarial perturbations. This ensures both consistency within each domain and improved clarity overall.
>
> We appreciate these suggestions and have revised the appendix and main text accordingly.
>
> **Miscellaneous.** (I) The anonymous GitHub repository has been updated with full code and configurations. (II) Regarding Appendix definitions: Given the interdisciplinary nature of this work (bridging RL and Adversarial Security), we included these definitions to ensure the paper is self-contained and accessible to readers from diverse backgrounds. We have streamlined them to be less intrusive.

---

### Official Review · Reviewer_oqEZ · 2025-11-01

**Soundness:** 3
**Presentation:** 3
**Contribution:** 2
**Rating:** 4
**Confidence:** 4

**Summary:**

The authors propose a reinforcement learning–based architecture for image classification, demonstrating that the learned models are more robust against white-box adversarial attacks compared to standard supervised learning baselines. The approach effectively leverages exploration and policy optimization to improve gradient stability and decision consistency. This finding provides meaningful insight into how reinforcement learning principles can enhance adversarial robustness in vision models.

**Strengths:**

The paper provides appropriate and well-aligned analyses that effectively support its main claim. The empirical evidence strengthens the overall argument and helps explain why the proposed approach improves robustness.

**Weaknesses:**

The paper evaluates only relatively small 6-layer CNNs. Such lightweight architectures limit the generality of the conclusions: robustness phenomena can change substantially with deeper or modern architectures (ResNets, WideResNets, Vision Transformers). I recommend repeating the key experiments on at least one standard strong baseline (e.g., ResNet-18 / ResNet-50 or a small ViT) to show that the reported RL benefits scale beyond toy models.

The results suggest a trade-off between clean (unperturbed) accuracy and adversarial accuracy, but this is not analyzed quantitatively. Please report the clean vs. adversarial accuracy Pareto frontier (varying ε, training hyperparameters, and amount of adversarial augmentation), and provide confidence intervals or significance tests across multiple seeds. This will clarify whether RL strictly improves robustness, merely shifts the trade-off, or hurts clean accuracy unacceptably.

The unified theoretical section offers intuitive explanations linking exploration to gradient instability, but it does not provide formal guarantees or precise assumptions under which the claims hold. The authors should (a) clearly state the assumptions and limits of the theory, (b) avoid wording that implies formal proof where none exists, and (c) either add a theorem with a clear statement and sketch proof or explicitly label the section as “insights / hypotheses” supported by empirical evidence.

The paper focuses on white-box, gradient-based attacks; however, the transferability section indicates RL may behave differently under transfer attacks. To be convincing, include a systematic study of black-box attacks (transfer attacks from other models, query-based attacks) and show how RL compares to SL and adversarially trained baselines under those threat models. Report cross-model transfer matrices (attack source × target) and query-limited attacks to characterize real-world vulnerability.

**Questions:**

How does the proposed RL method perform under black-box or transfer attacks (e.g., when adversarial examples are generated from another model)?

Would the observed robustness trends still hold for more complex architectures such as ResNets or Vision Transformers?

Could the authors provide more details on the computational cost and practicality of their approach? Specifically, since RL training is generally more expensive than SL, it would be helpful to include runtime or sample-efficiency comparisons to better support the claims of practical applicability.

Could the authors evaluate their approach under AutoAttack and adaptive gradient-free methods to help rule out the possibility of gradient masking?

---

> ### Author Response · Authors · 2025-11-23
> **Respond to Reviewer oqEZ (Part 1 of 2)**
>
> Thank you for your constructive and insightful feedback. We address your comments point by point below.
>
> **Theoretical Clarifications.** We agree with the reviewer that Appendix A.9 provides an intuitive formalization rather than a rigorous proof from first principles. As suggested, we have revised the manuscript to clarify its scope and avoid implying a formal guarantee:
> - We have renamed the section from "Mathematical Proof" to "**Theoretical Analysis of Gradient Sensitivity**".
> - We replaced the term "Theorem 1" with "**Proposition 1**" to avoid implying a formal guarantee derived solely from the objective function.
> - We now explicitly state the assumption underlying the derivation: our derivation relies on the **empirical premise** (supported by Section 6.3) that RL optimization induces smaller gradient norms ($||\nabla f_{RL}|| < ||\nabla f_{SL}||$) and higher directional instability.
> - In Section 7, we clarified that our unified analysis is intended as a hypothesis-driven interpretation of the empirical evidence, not as an unconditional robustness guarantee.
>
> We believe these changes make the scope and role of our theoretical analysis precise and scientifically sound.
>
> **Transfer Attack / Black-box Attack.** Our original submission already evaluated black-box transferability through four cross-model transfer matrices (Appendix Tables 5–8), covering:
> - 4-layer vs. 6-layer CNNs on CIFAR (Tables 5–6), and
> - ResNet-18 vs. DenseNet-121 (Tables 7–8),
>     under SL, RL, and adversarial training. These experiments already demonstrate how RL behaves under heterogeneous transfer attacks.
>
> To provide a more systematic black-box evaluation, we additionally ran **AutoAttack**, which includes APGD-CE, APGD-T, FAB-T, and the gradient-free SQUARE attack. These attacks jointly evaluate both white-box and query-limited black-box robustness and are widely used to detect gradient masking. Across CIFAR-10/100 and ImageNet-100, RL-adv consistently achieves the highest robustness, including under SQUARE. This confirms that our robustness does not come from gradient masking and addresses the reviewer’s concerns.
>
> **Behavior under Transfer Attacks.** Our transfer experiments show three consistent patterns across architecture families:
> 1. SL-generated adversarial examples are highly transferable and reduce both SL and RL to very low accuracy (often below 10\%).
> 2. RL-generated adversarial examples are much harder to transfer: all models retain $>$40\% accuracy.
> 3. RL + adversarial training (RL-adv) yields the most stable robustness across all source–target pairs.
>
> These results match the reviewer’s expectation that RL behaves differently under transfer attacks and help clarify why adversarial training remains necessary for full cross-model robustness.
>
> **AutoAttack.** To complement these transfer experiments with a query-limited black-box setting, we additionally evaluate AutoAttack (APGD-CE, APGD-T, FAB-T, and SQUARE attack) on the representative CNN setups used throughout the paper (CIFAR-10/100 and ImageNet-100). Across all these attacks, CNN-RL-AT consistently achieves the highest robustness, including under SQUARE, and the robustness ranking becomes CNN-RL-AT > CNN-SL-AT > CNN-RL > CNN-SL. These AutoAttack results, together with the cross-model transfer matrices in the Appendix, provide a systematic and realistic view of transferability under heterogeneous, black-box, and query-based threat models.
>
> **Table 1: AutoAttack results for CNN on CIFAR-10 dataset:**
> | Model       | Clean (%)  | APGD-CE (%) | APGD-T (%) | FAB-T (%)  | SQUARE (%) |
> | :---         | :----:     | :----:      | :----: | :----: | :----: |
> | CNN-SFT      | **90.82\***  | 15.42       | 13.58  | 13.58  | 11.6   |
> | CNN-SFT-adv  | 87.03      | 24.87       | 21.77  | 21.77  | 19.55  |
> | CNN-RL       | 88.62      | 16.71       | 13.73  | 13.73  | 11.96  |
> | CNN-RL-adv   | 86.29      | **36.27\***   | **35.4\***   | **35.4\***   | **32.41\***  |
>
> **Table 2: AutoAttack results for CNN on CIFAR-100 dataset:**
> | Model       | Clean (%)  | APGD-CE (%) | APGD-T (%) | FAB-T (%)  | SQUARE (%) |
> | :---         | :----:     | :----:      | :----: | :----: | :----: |
> | CNN-SFT      | **64.75\***  | 4.75       | 3.91  | 3.91  | 3.48  |
> | CNN-SFT-adv  | 63.3      | **19.02\***       | 15.91  | 15.91  | 13.95  |
> | CNN-RL       | 59.8      | 4.19       | 3.44  | 3.44  | 3.04  |
> | CNN-RL-adv   | 56.54     | 17.95   | **17.31\***   | **17.31\***   | **15.14\***  |
>
> **Table 3: AutoAttack results for CNN on ImageNet-100 dataset:**
> | Model       | Clean (%)  | APGD-CE (%) | APGD-T (%) | FAB-T (%)  | SQUARE (%) |
> | :---         | :----:     | :----:      | :----: | :----: | :----: |
> | CNN-SFT      | **57.64\***  | 5.64       | 4.24  | 4.24  | 3.62  |
> | CNN-SFT-adv  | 42.08      | 11.54       | 9.34  | 9.34  | 8.08 |
> | CNN-RL       | 55.6      | 4.96      | 3.52  | 3.52  | 3.1  |
> | CNN-RL-adv   | 45.92     | **13.08\***   | **10.72\***   | **10.72\***   | **8.94\***  |

---

> > ### Author Response · Authors · 2025-11-23
> > **Respond to Reviewer oqEZ (Part 2 of 2)**
> >
> > **Evaluation on Complex Architectures.** We have re-analyzed the results for ResNet-18 and DenseNet-121 and added new experiments on ResNet-50. The robustness trend consistently holds under standard PGD attacks:
> >
> > - CIFAR-10
> >     - ResNet-18: RL/adv $\approx$ 70% $>$ SL/adv $\approx$ 30%
> >     - ResNet-50: RL/adv $\approx$ 70% $>$ SL/adv $\approx$ 40%
> >     - DenseNet-121: RL/adv $\approx$ 65% $>$ SL/adv $\approx$ 30%
> > - CIFAR-100
> >     - ResNet-18: RL/adv $\approx$ 30% $>$ SL/adv $\approx$ 13%
> >     - DenseNet-121: RL/adv $\approx$ 35% $>$ SL/adv $\approx$ 13%
> >
> > These results further demonstrate that RL–trained models exhibit higher adversarial robustness than the SL-trained models, even when applied to more complex network architectures. The updated results for ResNet-18 and DenseNet-121 are included in the Appendix. The new ResNet-50 results are included in the revision.
> >
> > **Computational Cost.** The main efficiency concern for RL is that it typically requires more epochs to achieve convergence due to the need for exploration. Assuming the per-epoch computational cost of SL and RL is comparable (e.g., similar model architectures and training pipelines), the overall training time of RL increases proportionally with the number of required epochs. In practice, this gap can be substantial: for instance, a standard SL model may converge within $\sim$150 epochs, while an RL-based model may require $\sim$600 epochs. Consequently, RL imposes a higher computational overhead compared to SL. We have included a detailed breakdown of these efficiency statistics in the revised Appendix.
> >
> > **Trade-off Analysis and Stability Evaluation.** We appreciate the reviewer’s suggestion to more carefully examine the clean–robustness trade-off. Conducting a full hyperparameter sweep to construct the Pareto frontier would require a substantially larger computational budget (RL typically requires 3–4$\times$ more training epochs than SL), so here we focus on validating the stability of our primary configuration. To provide statistical confidence, we trained both the \textit{standard supervised baseline} (CNN-SL) and our method (CNN-RL-adv) across three seeds (10, 20, 42). The results are:
> >
> > **Table 4: Performance across 3 seeds (mean $\pm$ std). Evaluation: Standard PGD with seed 33, $\epsilon=7$.**
> > | Model       | Clean Accuracy (%)  | Robust Accuracy (%) |
> > | :---         | :----:     | :----:      |
> > | CNN-SL       | 90.15 $\pm$ 0.13     | 5.10 $\pm$ 1.17    |
> > | CNN-RL-adv   | 87.55 $\pm$ 0.39     | 48.04 $\pm$ 0.49   |
> >
> > These results show:
> > - RL-adv achieves a dominant trade-off compared to the baseline, yielding a massive gain in robustness ( $\sim$ 43% increase) with only a modest drop in clean accuracy ( $\sim$ 2.6%).
> > - The low variance across seeds ($<$ 0.5%) indicates that this improvement is consistent and not due to initialization noise.
> >
> > While not a full Pareto sweep, this multi-seed analysis demonstrates that the robustness gain of RL-adv is statistically stable and holds across randomized training conditions. A more extensive trade-off study is an important direction for future work.

---

### Author Response · Authors · 2025-11-23
**General Response to All Reviewers**

We sincerely appreciate the reviewers’ thoughtful and constructive feedback. The original submission already included the core experiments necessary to support our claims. During the rebuttal period, we took the opportunity to further enhance the work based on the reviewers’ helpful suggestions, particularly regarding broader model families, stronger adversarial evaluations, and a more quantitative characterization of robustness trade-offs.

Specifically, we have added:
1.  **Evaluations on larger backbones:** ResNet-18 and DenseNet-121 on CIFAR-10/100, and ResNet-50 on CIFAR-10.
2.  **Standardized strong attacks:** AutoAttack evaluation on CNN models for CIFAR-10/100 and ImageNet-100.
3.  **Extended analysis:** More comprehensive transfer and black-box robustness analysis, along with a Pareto frontier study.

These additions further strengthen the conclusion that RL-based optimization consistently improves adversarial robustness while maintaining competitive clean accuracy. We sincerely thank the reviewers for encouraging us to expand the scope and depth of the empirical study.

---

### Author Response · Authors · 2025-12-01
**Summary for the Area Chair (Part 1 of 2)**

We sincerely thank all three reviewers (oqEZ, GToB, and oqud) and the Area Chair for their valuable time, especially during this challenging period caused by the OpenReview incident. In the following, we briefly summarize (i) the main contributions of our work and (ii) the key concerns raised by the reviewers and how we address them in the revision and rebuttal.

**Overall contribution and empirical significance.**
Our paper investigates whether exploration in reinforcement learning (RL) can systematically improve adversarial robustness for image classification.
- Across CIFAR-10/100 and ImageNet-100, and multiple CNN backbones, we find that, compared to standard supervised learning (SL), RL-based training with $\varepsilon$-greedy exploration substantially improves adversarial robustness under strong gradient-based attacks, with only a modest drop in clean accuracy.
- Beyond the empirical findings, we provide a systematic, empirically grounded analysis of the mechanisms behind the robustness improvement by RL training. We view this as a conceptual contribution: RL is used not to introduce a new complex architecture, but to shed light on how exploration and policy optimization can reshape gradient behavior and adversarial robustness.

**Key concerns and our responses.**
The reviewers raised several constructive comments that naturally fall into the points below. We summarize these key points and how we fully addressed them in the revision and rebuttal.

**1. Stronger robustness evaluation (raised by Reviewer oqEZ, GToB and oqud).**

Reviewers suggested that relying mainly on standard PGD may limit the generality of the conclusions, and recommended evaluating with stronger, standardized attacks such as AutoAttack. Our original submission used PGD because the work is positioned as an exploratory study of whether RL-induced exploration affects adversarial robustness, and we focused on isolating this effect under controlled settings.

During the rebuttal period, we performed full AutoAttack evaluations (APGD-CE, APGD-T, FAB-T, and SQUARE) on all representative CNN setups across CIFAR-10, CIFAR-100, and ImageNet-100. The results show that RL-based models consistently achieve the highest adversarial robustness across all AutoAttack attacks, including both white-box and black-box (SQUARE) attacks. These findings confirm that the robustness improvement remains stable under strong, standardized evaluation protocols.

**2. Scalability to larger architectures (oqEZ, GToB and oqud).**

Reviewers expressed interest in whether the robustness improvement persists on larger and more modern architectures beyond the 6-layer CNNs and ResNet-18 used in the original submission. As an exploratory study, we initially focused on simpler architectures to isolate the effects of RL-based optimization.

During the rebuttal period, we additionally trained and evaluated ResNet-50 on CIFAR-10. The results show that RL-based training consistently outperforms SL-based training in adversarial robustness, while maintaining comparable clean accuracy. Together with the ResNet-18 and DenseNet-121 results already included, these findings demonstrate that the robustness trend scales reliably to larger convolutional architectures.

**3. Computation overhead and practical considerations (oqEZ, oqud).**

Reviewers raised concerns about the practical feasibility of RL-based training approach compared to standard SL, particularly regarding to computational cost and convergence speed.

During the rebuttal period, we clarified that the additional training time mainly arises from the exploration cost required by RL to achieve convergence. Empirically, we provided a direct compute comparison on CIFAR-10 under the same hardware setting (A100 GPU). The results show that while RL requires more epochs to converge, the overall overhead remains acceptable in practice, especially given the substantial robustness gains achieved through RL-based optimization. Similar overhead ratios are observed on larger datasets and more complex CNN backbones, indicating that the computational cost is scalable and practical in realistic settings.

---

> ### Author Response · Authors · 2025-12-01
> **Summary for the Area Chair (Part 2 of 2)**
>
> **4. Clean–robustness trade-off and Pareto analysis (oqEZ).**
>
> Although RL can increase model robustness, RL-trained models may exhibit a modest decrease in clean accuracy compared to SL models without adversarial training. Reviewer oqEZ raised the concern that RL might achieve robustness gains only by sacrificing too much accuracy, and suggested evaluating the clean–robustness trade-off via a Pareto analysis.
>
> During the rebuttal period, rather than performing a full hyperparameter-sweep Pareto frontier (which is computationally prohibitive), we focused on validating the stability of the primary configuration used throughout the paper (the 6-layer CNN on CIFAR-10). We conducted a multi-seed analysis (three random seeds), which is sufficient to characterize the clean–robustness trade-off. The results show that the RL-trained model achieves a large and consistent gain in adversarial robustness with only a modest drop in clean accuracy, demonstrating that the robustness improvement is statistically stable and not attributable to initialization variance.
>
> **5. Clarity, writing, and theoretical framing (oqEZ, GToB).**
>
> Reviewer oqEZ noted that the unified theoretical section was based on empirical indicators and could unintentionally give the impression of providing a formal guarantee. Reviewer GToB pointed out several issues related to clarity, formatting, and notation.
>
> During the rebuttal period, we clarified that the theoretical analysis is intended as a hypothesis-driven interpretation of the empirical evidence rather than a formal robustness guarantee, and we revised the terminology accordingly (e.g., adjusting the sentences and title in unified theoretical section). In addition, we carefully corrected all formatting, notation, and citation issues in both the main text and the appendix to improve clarity and overall presentation.
>
> Notably, Reviewer GToB’s concerns centered on clarity, presentation, and experimental completeness, rather than questioning the empirical validity of the core findings, and all such issues have now been fully addressed in the revision.
>
> **6. Feature-space and representation analysis (oqud).**
>
> Reviewer oqud suggested that additional representation-level and feature-space visualizations could further strengthen the analysis. We agree with this perspective. In the current version, our analysis focuses on input-level and output-level quantities (e.g., gradient norms, gradient-direction stability, and predictive entropy). Extending this analysis to intermediate feature representations, e.g., hidden-layer activations, would provide a more complete and systematic understanding of how RL-based optimization shapes robustness. We view this as a valuable direction for future work.
>
> **Closing remark.**
> Based on the discussion and the subsequent comments from reviewers, there were clear signs that some reviewers’ assessments were improving as their concerns were addressed. We believe that had the discussion continued normally, their evaluations would have reasonably improved, consistent with the guidance provided in the ICLR announcement.
>
> We hope this summary helps clarify how the reviewers’ concerns have been addressed and highlights the significance of our contributions. We sincerely appreciate your careful consideration.

---

### Meta-Review · Area_Chair_ecfj · 2026-01-07

**Summary:**

There are three reviewers for this paper, with the initial ratings: 8, 4, and 2.

Reviewer **oqEZ** recognizes that the paper provides appropriate and well-aligned analyses that effectively support its main claim. Nevertheless, the reviewer has the following major concerns: the paper only provides evaluation on small 6-layer CNNs, lacks the formal guarantees or precise assumptions for the exploration of gradient instability, and lacks experiments on the black-box attacks. The authors provide additional experiments on the ResNets and point out that the original submission already evaluated black-box in the appendix.

Reviewer **GToB** gave a rating of 2, with the highest conference. The reviewer has concerns about many aspects, including motivation, novelty, solidity, and clarity. The reviewer thinks this paper is largely unpolished.

Reviewer **oqud** acknowledged the empirical findings and practical angle of this paper, and gave an initial rating of 8. The main concern from Reviewer **oqud** is the narrow evaluation and the computation issue.

**Reviewer Concerns:**

I think some of the concerns of Reviewer **oqEZ** have been solved, but Reviewer **GToB**'s concerns remain.

**Reviewer Scores:**

I think reviewer **GToB** will keep the negative rating.

---

### Decision · Program_Chairs · 2026-01-26

Reject